# RETHINKING ARTISTIC COPYRIGHT INFRINGEMENTS IN THE ERA OF TEXT-TO-IMAGE GENERATIVE MODELS

**Mazda Moayeri**[1]**, Sriram Balasubramanian**[1]**, Samyadeep Basu**[1]**,**

**Priyatham Kattakinda**[1]**, Atoosa Chegini**[1]**, Robert Brauneis**[2]**, Soheil Feizi**[1]
[1]University of Maryland, [2] George Washington University

## ABSTRACT

The advent of text-to-image generative models has led artists to worry that their individual styles may be copied, creating a pressing need to reconsider the lack of protection for artistic styles under copyright law. This requires answering challenging questions, like what defines style and what constitutes style infringment. In this work, we build on prior legal scholarship to develop an automatic and interpretable framework to *quantitatively* assess style infringement. Our methods hinge on a simple logical argument: if an artist's works can consistently be recognized as their own, then they have a unique style. Based on this argument, we introduce `ArtSavant`, a practical (i.e., efficient and easy to understand) tool to (i) determine the unique style of an artist by comparing it to a reference corpus of works from hundreds of artists, and (ii) recognize if the identified style reappears in generated images. We then apply `ArtSavant` in an empirical study to quantify the prevalence of artistic style copying across 3 popular text-to-image generative models, finding that under simple prompting, 20% of 372 prolific artists studied appear to have their styles be at risk of copying by today's generative models. Our findings show that prior legal arguments can be operationalized in quantitative ways, towards more nuanced examination of the issue of artistic style infringements.

## 1 INTRODUCTION

Currently, US copyright law offers no protection for artistic styles, likely due to challenges in defining style infringement and a prior lack of necessity. However, with highly capable text-to-image generative models transforming the art landscape, there have been increasingly more calls to reconsider this lack of protection. Namely, many fear that models like Stable Diffusion, Imagen, Mid-Journey, and DeepFloyd (Rombach et al., 2021; Saharia et al., 2022; DeepFloyd, 2023; Podell et al., 2024) may make replication of an artist's unique style as simple as providing an adequate prompt, potentially inundating the market with imitations that devalue the original artist's work and threaten their livelihood. This matter has gained widespread attention, as it is fundamentally interdisciplinary (engaging legal, artistic, and technical communities), and has serious material and human consequences. In fact, multiple legal cases regarding artistic copyright are ongoing (Brittain, 2024; Poritz, 2024), and Adobe (2023) has already called for new provisions in copyright law to protect artistic style. Thus, with generative AI creating a pressing need to answer previously unsettled questions, it is now critical for us to rethink artistic copyright.

We begin by surveying existing copyright law around art, finding that it has historically relied heavily on qualitative judgments, such as the 'substantial similarity test' (Goldstein, 2014). The subjective nature of existing law has led to judgments that vary from case-to-case (e.g. on if characters are protected (DCC, 2015; MGM, 1995; Kli, 2014) or not (Sony, 1998)), which will prove problematic when dealing with AI-powered mimicry at unprecedented scale. An automatic quantitative approach could help make judgments quicker and more consistent. However, any technical solution *must* also be easy to understand and based in legal literature, as the audience whose decisions we intend to assist[1] is largely non-technical (e.g. judges, juries, and artists). Thus, we set out to develop an intuitive, automatic, and legally-grounded manner to quantitatively argue artistic style infringement.

---

[1]We absolutely do not wish to replace humans in this nuanced decision making, but instead to help them.

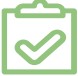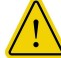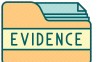

**ArtSavant Report for *Canaletto***

**You have a unique and recognizable style!**
We can identify your style (over the style of 372 other artists) in **88.37%** of your works. This puts you in the top 83.6% percentile of artists in recognizability.

**Your style is detected in works generated by Stable Diffusion.**
When prompting a gen AI model to copy you, the resultant images exhibit your style more than 372 other artists **70.34%** of the time.

**We find stylistic elements unique to you that reappear in generated images.**
We identify some tag signatures (set of stylistic elements that frequently co-occur only in your work) that also appear in generated images. Here's an example; click to see more.

Matched Tag for Canaletto: oil painting, broad brushwork, geographical symbolism
0 other artists also have this signature

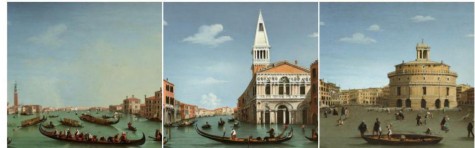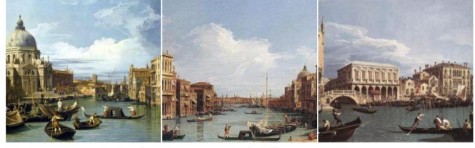

17 generated images with this signature          8 real images with this signature

Figure 1: Our primary contribution is an interpretable and quantitative framework, rooted in legal scholarship, for arguing style infringement from the perspective of classification. Given artworks by Canaletto, our tool, `ArtSavant`, automatically identifies a unique style and recognizes said style in generated art, summarizing its findings in an easy to understand yet quantitative report.

Our work is structured around key questions we identify that, when answered, can form the basis of an argument for artistic style infringement. Namely, given an artist, **i.** do they have a unique style? If so, **ii.** to what degree does it appear in generated art? And lastly, **iii.** in what precise way (with respect to specific artistic elements present in both the original and generated art) is the style copied? Inspired by legal arguments presented by Sobel (2024) and O'Connor (2022), we first frame artistic style as characterized by a set of elements that co-occur frequently across an artist's *body of work*, where each element may not be protectable on its own, but together, they could represent an unique stylistic *signature*.

To prove the *uniqueness* of an artistic style, we propose a simple logical argument, which serves as our core ideological contribution. Namely, if an artist's works are consistently recognized as their own, that artist must have some unique style, with which their work can consistently be classified back to them (instead of hundreds of other potential creators). Therefore, the task of showing the existence and uniqueness of artistic styles can be studied *from the lens of classification* – something deep networks are particularly adept at doing. To perform this classification, we curate a reference dataset of artworks from 372 artists, and implement two classification methods, taking holistic and analytic approaches.

Current copyright law dictates that similarity between artworks must be evaluated in *analytic* and *holistic* terms (Tuf, 2003; Goldstein, 2014) (see § 2). For example, we could *analyze* Vincent Van Gogh's style as comprised of expressive wavy lines, bright unblended coloring, post-impressionism, choppy textured brushwork, etc, or we could make an intuitive, *holistic* judgement: e.g. in Figure 3, we can tell the look and feel of the generated images capture Van Gogh's style, even without articulating the shared stylistic elements. In order to make these notions more concrete and quantitative, we develop two complementary methods corresponding to holistic and analytic ways of evaluating style similarity.

The first method – *DeepMatch* – is simply a neural network classifier, with which we show that for 89.2% of artists, their held-out works can consistently (i.e. over half the time) be mapped back to them (over 371 other artists). This represent a significant empirical finding, as it quantitatively shows that **unique artistic styles exist for an overwhelming majority of artists** we study – a necessary precondition for adding legal protections for artistic style. DeepMatch can be thought of as capturing *neural signatures* for each artist. Namely, the classification head vector consists of a combination of neural features that encode that artist's style. As these are not interpretable, we complement the *holistic* DeepMatch with *TagMatch*, a novel, inherently interpretable, attributable, *analytic* approach.

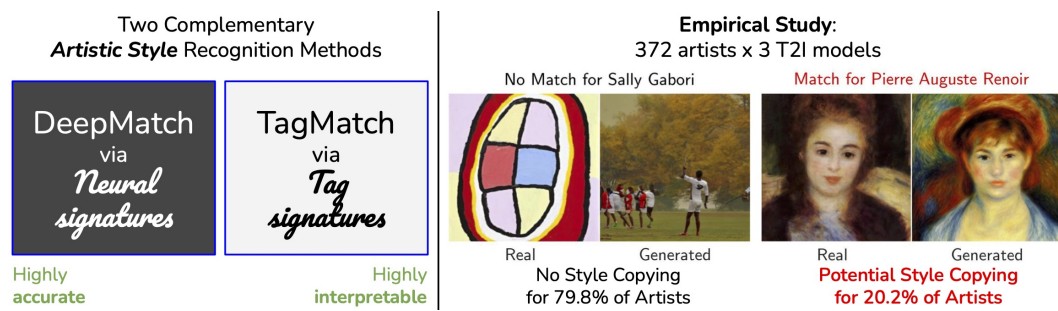

Figure 2: We define artistic style as a set of elements (or signature) that appear frequently over a body of work, and reduce the problem of style copy detection to classification of *sets* of images to artists. (**left**) We offer proof-of-concept via two ways to recognize artistic styles over image *set*, including a novel inherently interpretable and attributable tag-based method. (**right**) In an empirical study of 372 prolific artists, we find generative models potentially copy artistic styles for 20.2% of these artists.

TagMatch maps a collection of works to an artist by first assigning diverse stylistic tags in a zero-shot manner to each work, and then efficiently searching over the space of tag compositions to surface unique *tag signatures*. We show empirically that searching over tag compositions is critical, as no tag on its own is unique to an artist – recall this idea was described by legal scholars Sobel (2024) and O'Connor (2022); in our work, we make this quantitative. Further, importantly, **TagMatch allows for articulating the specific combination of elements that uniquely appear in artist's works and in (infringing) generated work**. Such interpretable output can be particularly useful for the legal settings where arguments around artistic copying ultimately are to be made.

We package these methods in `ArtSavant`, a tool that for any artist can produce a report like Figure 1 in minutes. `ArtSavant` shows our logical framework to argue style infringement is practically feasible, automatically generating a quantitative assessment of style copying in a way that can be understood by the broad set of relevant stakeholders. Finally, we use `ArtSavant` to provide a quantitative snapshot of the prevalence of style copying with today's generative models under simple prompting. After generating images in the style of artists from our dataset with 3 popular text-to-image models and using `ArtSavant` to assess copying, we find 20% of the artists we study to be at risk, suggesting style copying may indeed by possible with today's generative AI.

In summary, we make the following contributions:

- We develop an automatic, interpretable, and **legally-grounded manner to quantitatively argue artistic style infringement**, leveraging a simple classification argument.

- We introduce `ArtSavant`, **a practical tool to quantify and communicate style copying for any artist**, consisting of a reference dataset of artworks from 372 prolific artists, and two complementary methods, including a novel, highly interpretable and attributable one. Notably, our two methods directly relate to the principles of "analytic" and "holistic" style similarity used in the substantial similarity test in existing copyright law.

- With `ArtSavant`, we perform **a large-scale empirical study** to measure style copying across 3 popular text-to-image generative models, finding that generated images (using simple prompting) from 20% of the artists examined appear to be at risk of style copying.

## 2 RELATED WORKS

**Overview of copyright legal literature:** Currently, there is an ongoing debate within legal scholarship[2] regarding whether artistic style can and should be protected under copyright law. Some (Bracha, 2023) conclude, from a traditional interpretation of copyright law, that style is wholly outside the purview of copyright law as copyright is only concerned with protection against copying *individual* artworks. However, many courts have recognized protection for characters that are developed over

---

[2]We encourage readers to review Appendix B for a detailed but brief history of copyright law, so to understand the broader context for our work and the legal challenges / ambiguities surrounding it.

multiple works, from Sherlock Holmes (Kli, 2014) to James Bond (MGM, 1995) and the Batmobile (DCC, 2015). Other scholars (O'Connor, 2022; Sobel, 2024) have argued that artistic style, separate from any particular artwork, constitutes a distinct intellectual contribution which has commercial and aesthetic value. In particular, Sobel (2024) suggests that style can be defined as "a holistic attribute of a work, or a group of works, that comprises a constellation of expressive choices. These expressive choices might be unprotectable individually, but in combination, they may constitute protectable expression." Additionally, Sobel emphasizes the challenge courts face when applying the *substantial similarity test* to art – they "must simultaneously dissect images into their constituent elements — a task judges claim they are unable to do— while also assessing works' aesthetic appeal holistically and intuitively". If indeed style may come under copyright protection in the future, a substantial similarity test for art style would be even more challenging to implement and apply than one for individual artworks. In our work, we aim to develop a tool which can help courts determine the extent of style similarity from the lens of the substantial similarity test applied to artistic style. Critically, we do not seek to replace human decision makers with an automated analog, but instead to leverage our tool to provide quantitative evidence to assist humans in making nuanced qualitative judgments.

**Current technical solutions:** The rapid advance of image generative models has made the possibility of mimicking artists' personal styles a topic of discussion in the literature (Ren et al., 2024). Some works describe ways to either detect direct image copying in generated images, or to foil any future copying attempts by imperceptibly altering the artists' works to prevent effective training by the generative models. These include techniques like adding imperceptible watermarks to copyrighted artworks (Wang et al., 2024; Cui et al., 2023; 2024), and crafting "un-learnable" examples on which models struggle to learn the style-relevant information (Shan et al., 2023; Xue et al., 2024; Zhao et al., 2023). Others have suggested methods to mitigate this issue from the model owner's perspective - to either de-duplicate the dataset before training (Carlini et al., 2023; Somepalli et al., 2022; 2023), or to remove concepts from the model after training ("unlearning") (Kumari et al., 2023; Gandikota et al., 2023; Basu et al., 2023). Methods like (Carlini et al., 2023; Somepalli et al., 2022; 2023) are also more focused on analyzing direct image copying from the training data, and thus may not be applicable to preventing style copying.

**Shortcomings of current methods from a legal perspective:** None of these works tackle the problem of *detecting* potentially copied art *styles* in generated art, especially in a manner which may be relevant to legal standards of copyright infringement. According to current US legal standards (CRS, 2023), an artwork has to meet the "substantial similarity" test for it to be infringing on copyright. This similarity has to be established on *analytic* and *holistic* terms (Tuf, 2003; Goldstein, 2014). Analytic here refers to explaining an artwork by breaking it down into its constituents using a concrete and objective technical vocabulary, while holistic refers to the overall "look and feel" of the artwork. So to be relevant to the legal community (who ultimately decides on alleged cases of style copying), we design our tool to reflect this dichotomy in its working, while also emphasizing ease of use and interpretability, to make our tool practically useful for a concerned artist hoping to protect themselves. These priorities manifest in our reformulation of detecting style copying as classification in §4. But first, we discuss limitations in applying the typical copy detection approach to artistic styles.

## 3 MOTIVATION: IMAGE-WISE SIMILARITY MAY BE LIMITED FOR STYLE COPYING

A prevailing approach to investigating copying involves representing images in a deep embedding space via models like SSCD (Pizzi et al., 2022) or DINO (Caron et al., 2021a), and computing image-to-image similarities across generated and real images. Such an approach has been employed by Somepalli et al. (2022; 2023); Carlini et al. (2023) to show that generative models can (though rarely do) create exact replicas of training images. Inspired by these results and the consequent concerns from artists, we first explore if generative models can recreate famous artworks, e.g., by Vincent Van Gogh. Specifically, we generate images by prompting "*{artwork title}* by Vincent Van Gogh" for 1500 Van Gogh works, and compute the DINO similarity between pairs of a real and corresponding generated image. Figure 3 visualizes the distribution of similarities, as well as examples at each similarity level. We find that the vast majority of similarities are lower than 0.75, which amounts to pairs that are far from duplicates. However, even when the generated image differs significantly from the source real image, certain stylistic elements associated with Van Gogh seem to appear consistently in the generated works. Thus, **while instance-wise copying of artwork**

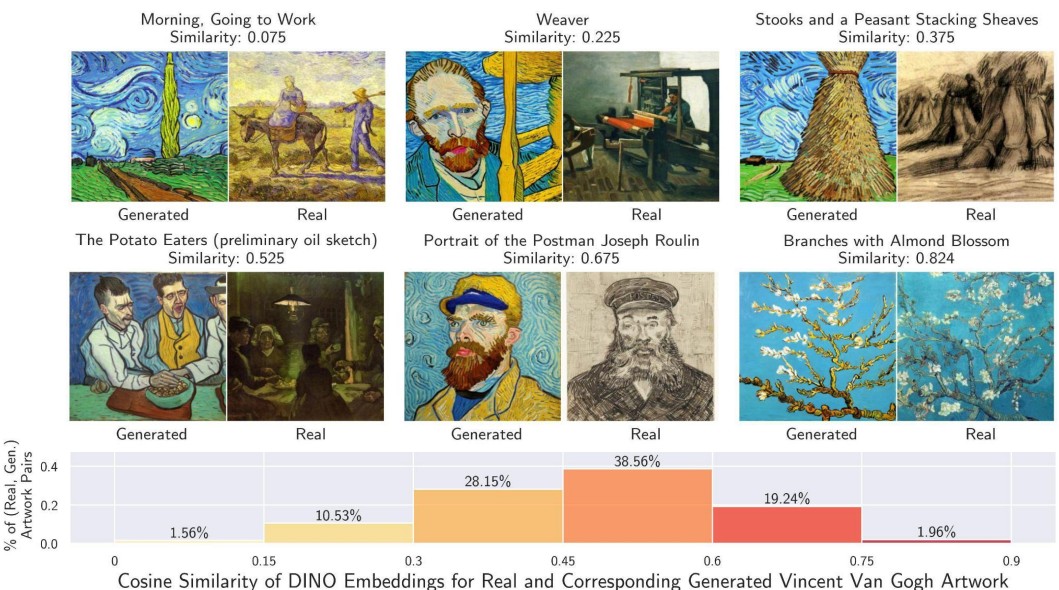

Figure 3: Example generations from Stable Diffusion 2 when prompted to produce specific paintings by Vincent Van Gogh, along with the histogram of similarities between the generated image and corresponding real image. Even for a famous artist like Van Gogh, generative models rarely produce near-exact duplicates. However, Van Gogh's *style* appears consistently, even when similarity is low.

**appears rare for even the ultra famous Van Gogh, style copying may require going beyond image-to-image comparisons**, as artists may still have their personal styles, developed over a long career/many artworks and at significant personal cost, infringed upon in ways that searching for exact replicas would miss. A recent work finetunes embeddings so that cosine similarity better proxies style similarity (Somepalli et al., 2024), though even in this case, the utility of such a tool in court is limited by its lack of interpretability. We provide a deeper comparison to this work in Appendix D.1.

## 4 REFORMULATING ARTISTIC STYLE COPYING AS CLASSIFICATION OVER IMAGE SETS

Having established that style is comprised over a body of work (instead of a single image) and that copy detection must be interpretable to hold weight in court, we now present an alternate framework for arguing style infringement, with the following intuition: if an artist's work can consistently be distinguished from that of other artists, then there must exist something unique that is present across that artist's portfolio. Thus, we can use classification over image sets (i.e. bodies of work) to demonstrate a unique style exists given an artist. Then, style infringement can be argued by showing the copied artist can again be predicted (over many others) given a set of generated works. We now detail DeepMatch and TagMatch, two complementary methods (w.r.t. accuracy and interpretability) that classify artistic styles over image sets, in holistic and analytic manners respectively.

**A necessary preliminary: WikiArt Dataset.** To distinguish one artist's style from that of others, we need a corpus of artistic styles (i.e. portfolios from many artists) to compare against. To this end, we curate a dataset $\mathcal{D}$ consisting of artworks from WikiArt [3] (like others (Tan et al., 2017; Karayev et al., 2014)) to serve as (i) a reference set of artistic styles, (ii) a validation set of real art to show (most) artists have unique styles and our methods can recognize them on held-out sets of their works, and (iii) a test-bed to explore if text-to-image models replicate the styles of the artists in our dataset in their generated images. We include ~91k artworks from 372 artists $\mathcal{A}$ spanning diverse eras and art movements, including any artist with at least 100 works on WikiArt. Each work is labeled with its genre (e.g., *landscape*) and style (e.g., *Impressionism*), though we primarily use the artist and title labels. We provide an easy-to-execute script to enable others to scrape newer versions of this dataset if desired. We now detail DeepMatch and TagMatch, which each compare a test set of images to our reference corpus.

---

[3]https://www.wikiart.org/; **note that we only include Public domain or fair use images**.

## 4.1 DEEPMATCH: BLACK-BOX DETECTOR

DeepMatch consists of a light-weight artist classifier[4] (on images) and a majority voting aggregation scheme to obatin one prediction for a *set* of images. Majority voting requires that at least half the images in a test set $\hat{D}$ are predicted to an artist $a$ for DeepMatch to predict $a$, allowing for abstention in case no specific style is recognized with sufficient confidence. For our classifier, we train a two layer MLP on top of embeddings from a frozen CLIP ViT-B\16 vision encoder (Radford et al., 2021), using a train split containing $80\%$ of our dataset. We employ weighted sampling to account for class imbalance. Since we utilize frozen embeddings, training takes only a few minutes on one RTX2080 GPU. Thus, a new artist could easily retrain a detector to include their works (and thus encode their artistic style).

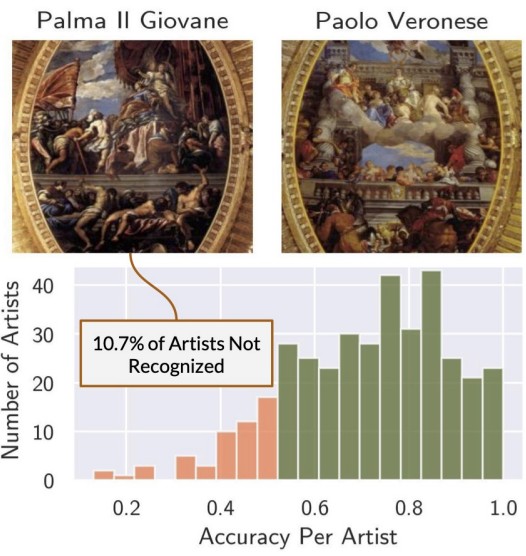

**Validation of the Detector.** We apply Deep-Match on the held-out test split of our dataset and observe that the image-wise classifier attains $72.8\%$ accuracy per image over $372$ artists. When aggregating image-wise predictions via majority vote, $89.3\%$ of artists are matched, validating our method, and **offering strong evidence towards the existence of unique artistic styles**. Specifically, neural classifiers capture unique and frequently co-occurring characteristics of the artists in their embedding space, which can be thought of as 'holistic' *neural signatures*. Figure 4 shows the distributions of image-wise accuracies per artist, shading correctly matched artists (green). We also present an image from one of the few artists who's style is not matched by DeepMatch, along with an image from a similar artist. Notice that the style

Figure 4: DeepMatch on held-out real art: $89.3\%$ of artists can be recognized. The remaining $10.7\%$ of artists have very similar styles to other artists: e.g., Palma Il Giovane's work differs marginally from other Italian renaissance painters.

of two artists can be extremely similar (see Appendix C.1), making the existence of unique artistic styles for the vast majority of artists considered (by way of neural signatures) a non-trivial observation. **This empirical finding also has legal significance, as it is necessary for an artist to demonstrate that they have a unique style before they could allege that their style is being infringed upon.**

## 4.2 INTERPRETABLE ARTISTIC SIGNATURES

Now we provide an analytic complement to DeepMatch's holistic approach. Namely, we seek to articulate the elements that comprise an artist's unique style. We do so by tagging images with descriptors (called atomic tags) drawn from a vocabulary of stylistic elements. Then, we *compose* tags efficiently to go from atomic tags that are common across artists to longer tag compositions that are unique to each artist (i.e. *tag signatures*). We detail these steps now, before explaining how tag signatures can be used to classify an image set to an artist in the following section.

**Zero-shot Art Tagging.** We utilize the zero-shot open-vocabulary recognition abilities of CLIP to tag images with descriptors of stylistic elements. First, we construct a concept vocabulary $\mathcal{V}$ with help from LLMs. Namely, we prompt Vicuna-13b and ChatGPT to generate a dictionary of concepts along various aspects of art. We manually consolidate and amend the concept dictionary, resulting in a vocabulary of 260 concepts over 16 aspects (see Appendix F.1).

To assign concepts to images, **we a design a novel zero-shot scheme that consists of selective multilabel classification per-aspect**. Namely, for an image, we compute CLIP similarities to all concepts, and normalize similarities *within each aspect*. Then, we only assign a concept its normalized

---

[4]Others have trained art classifiers (Karayev et al., 2014; Johnson et al., 2008; van Noord et al., 2015), but they do not operationalize them for style infringement.

''Guitar, Sheet music and Wine glass'' by Pablo Picasso

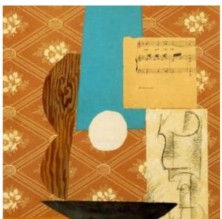

Tags:
abstract expressionism style
collage
repetitive composition
musical instruments
simple colors
Contemporary influences
abstract subject matter

''Leda Atomica'' by Salvador Dali

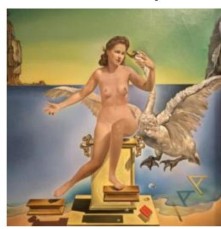

Tags:
magic realism style
heaven
Contemporary influences
surrealistic
sharp angles

Figure 5: Example atomic tags assigned via our proposed CLIP-based zero-shot method. We perform selective multilabel classification along various aspects of art (e.g. medium, colors, shapes, etc), so that atomic tags span diverse categories. Details in section 4.2.

similarity (i.e. z-score) exceeds a threshold of $1.75$. This means that a concept is only assigned for an aspect if the image is substantially more similar to this concept than other concepts describing the same aspect. Classifying per-aspect allows for a diversity of descriptors to emerge, while global thresholding results in a biased tag description, as concepts for certain aspects (e.g. subject matter) consistently have higher CLIP similarity than those for more nuanced aspects (e.g. brushwork). We call the assigned concepts *atomic tags*; figure 5 shows atomic tags assigned for a few examples.

**Validation of Quality of Tags Using Human-Study.** We validate the effectiveness of our tagging via a human-study involving MTurk workers. In particular, given an image of an artwork and an assigned atomic tag $v_{predict}$ from the vocabulary $\mathcal{V}$ – MTurk workers are asked "*Does the term $v_{predict}$ match (i.e. the concept $v_{predict}$ present) the artwork below?* ". The workers are then asked to select between {Yes, No, Unsure}. We collect responses for 1000 images with 3 annotators each. We find that in only $17\%$ cases, a majority of workers disagree with the provided tag, suggesting our tagging results in a low false positive rate. We also observe all three annotators agree in only $51\%$ of cases, reflecting that describing artistic style can be subjective. While our tagging is not perfect, it is a deterministic and automatic method of articulating artistic style elements, and it will improve as underlying VLMs improve too. See Appendix F.6 for more details and discussion on the human study.

**Tag Composition for Artists.** Using the atomic tags in the artwork specific vocabulary $\mathcal{V}$, we now present an iterative algorithm to obtain a set of *tag signatures* $\mathcal{S}_a$ for each artist $a \in \mathcal{A}$, which consist of multiple atomic tags that frequently *co-occur* in an artist's works. In particular, our algorithm efficiently searches the space of tag compositions to go from atomic tags to composition of tags which become more unique as the length of the tag composition grows. For e.g., while $40\%$ of the artists may use simple colors, *only* $15\%$ may use both simple colors and impressionism style *in the same works*.

To efficiently search the space of tag compositions per artist $a \in \mathcal{A}$, we first assign a set of tags to each of their images $x \in \mathcal{D}_a$ via the zero-shot *selective multi-label classification* method described above. For each image $x$, let tag$(x)$ denote the set of predicted atomic tags. To get atomic tags *for an artist*, we aggregate all atomic tags over images, and keep only the tags occurring in at least 3 works. We denote this aggregate set of atomic tags as the "Common Atomic Tags Per Artist" and denote it as $\mathcal{C}_a$. Then, we iterate through all the images $x \in \mathcal{D}_a$ for a given artist $a$, to find the intersection $I(x) = \text{tag}(x) \cap \mathcal{C}_a$. We then compute a powerset $\mathcal{P}(I(x))$ of the tags occurring in the intersection $I(x)$ and increment the count of each occurrence of the tag composition from the powerset in $\mathcal{S}_a$.

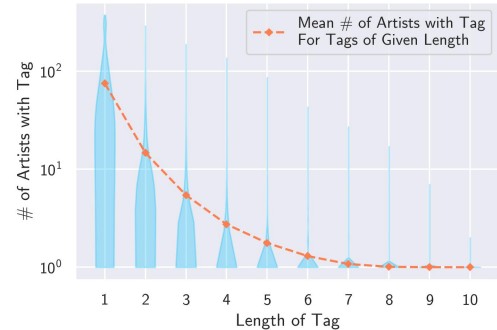

Figure 6: Composing atomic tags results in more unique tags, towards artistic *tag signatures*.

Note that the size of $I(x)$ is much smaller than that of $\mathcal{C}_a$, and thus, iterating through $\mathcal{P}(I(x))$ for each image $x$ is much, much faster than iterating through $\mathcal{P}(\mathcal{C}_a)$. Finally, we again filter the tag

compositions in $\mathcal{S}_a$, only including those that occur in at least 3 works. We provide the details of this tag composition algorithm in 1 and Appendix F.3.

**Can Unique Artistic Signatures be Articulated?** Applying tag composition on the WikiArt dataset, we observe that when the length of a tag (i.e. number of atomic tags that all frequently co-occur) grows, the number of artists that the tag (composition) is common for decreases (fig. 6). Eventually, our tag composition algorithm identifies unique tag *signatures* such that *only* one artist frequently produces art containing all atomic tags in the composition. This echoes the previous finding that unique styles exist, with these styles now articulated, towards an interpretable argument of style copying.

### 4.3 TagMatch: Interpretable and Attributable Style Detection

In 4.1, we outlined a holistic approach to accurately detect artistic styles. While DeepMatch obtains high accuracy (recognizing styles for $89.3\%$ of artists), the neural signatures it relies upon lack interpretability. For a copyright detection tool to be useful in practice (e.g., to be used as assistive technologies), providing explanations of the classification decisions can tremendously benefit the end-user. To this end, we leverage our efficient tag composition algorithm as defined in 4.2 to develop TagMatch - an interpretable classification and attribution method which can effectively classify a set of artworks to an artist, as well provide reasoning behind the classification and example images from both sets that present the matched tag signature. TagMatch follows the intuition of matching a test portfolio to a reference artist who's portfolio shares the most unique tag signatures. Given a set of $N$ test images $\mathcal{T} = \{x_i\}_{i=1}^{N}$, we first obtain a number of tag compositions for them using our iterative algorithm in 4.2. These tag compositions are then compared with the tag compositions of the artists in the reference corpus in order of uniqueness (i.e. we first consider tag signatures present in the test portfolio that occur for the fewest number of reference artists). We can then rank reference artists by how unique the shared tags are with the test portfolio. Detailed steps of the algorithm is in Appendix F.3. Notably, TagMatch is fast, taking only about a minute, after caching embeddings of all images.

**Validation of TagMatch.** We again utilize the test split of our WikiArt Dataset to validate the proposed style detection method. TagMatch predicts the correct artist with top-1 accuracy of $61.6\%$, with top-5 and top-10 accuracies rising to $82.5\%$ and $88.4\%$ respectively. While less accurate than DeepMatch, the *tag signatures* provided by TagMatch allow for analytic arguments to be made regarding style copying, as the exact tag signatures used in matching can be inspected. Moreover, the subset of images in both the test portfolio and matched reference portfolio can be easily retrieved, offering direct attribution of the method; examples can be seen in the next section, where we match generated images to our reference artists. Overall, we hope TagMatch and DeepMatch can serve as automatic and objective tools to navigate the subtle problem of identifying artistic styles, towards detecting style copying and helping artists argue their case (i.e. in a court of law) in such instances.

## 5 ArtSavant: A Practical Tool for Concerned Artists

We package DeepMatch and TagMatch into `ArtSavant`, a practical tool designed with a concerned artist in mind. Given a set of works by the concerned artist, `ArtSavant` would create an easy-to-understand report characterizing the degree to which generative models copy the styles of the artist. As shown in Figure 7, the artist can present a set of generated images, or we can generate them by prompting text-to-image models with prompts of the form "{title of work} by {name of artist}". The provided works are then combined with our existing art repository and split into train/test sets. Using the train split, we (a) train a classifier over the $372 + 1$ artists, and (b) tag all images, compose tags within artists, and store extracted tag compositions per artist, resulting in neural and tag signatures. With these, we can apply DeepMatch and TagMatch respectively. Applying DeepMatch to the held-out art provides a measure of recognizability, establishing that the artist has an identifiable style to begin with. Then, running DeepMatch on generated images provides a quantitative manner to understand if (and to what degree) the artist's style appears consistently in generated works. Finally, running TagMatch on the generated images helps articulate the particular style signatures that are copied, enabling an analytic way to argue infringement, while also surfacing stylistically similar examples.

Figure 1 shows an example report outputted by `ArtSavant` when presented with art from an artist named Canaletto, who we observed was at risk of style infringement. We design the report to be easy

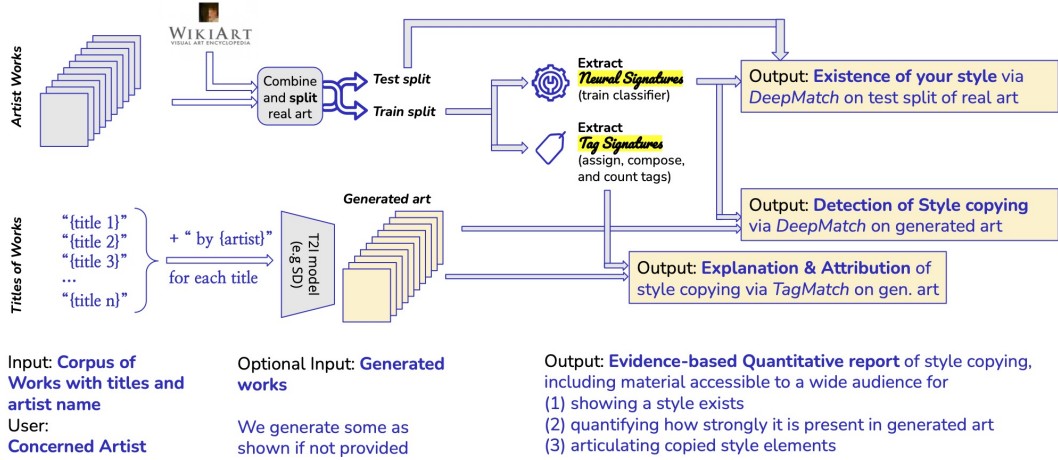

Figure 7: `ArtSavant` flow. We design our tool with a concerned artist in mind, who wishes to quickly investigate the degree to which they may be at risk of style copying by generative models.

to understand, quantitative, and legally-grounded. Moreover, the report can be generated very quickly: as all steps operate on embeddings from a frozen CLIP encoder, the process takes about 1-2 minutes.

## 5.1 ANALYSIS WITH ARTSAVANT: QUANTIFYING STYLE COPYING OF 372 PROLIFIC ARTISTS

While enough anecdotal instances of style mimicry have been observed to raise concern (Shan et al., 2023; Ren et al., 2024), the prevalence and nature of such instances remains nebulous. To shed quantitative insight on style copying, we now leverage `ArtSavant` on the 372 artists from our WikiArt dataset, generating images with three popular text-to-image models: (i) Stable-Diffusion-v1.4; (ii) Stable-Diffusion-v2.0; and (iii) OpenJourney from PromptHero. Following figure 7, we employ a simple prompting strategy of augmenting painting titles with the name of the artist; we explore alternate prompts in E.

We first apply **DeepMatch** to see what fraction of artists' styles can be recognized consistently over generated images. Namely, each generated image is classified to one of 372 artists, and per artist, predictions are aggregated via majority voting. Figure 8 shows the 'accuracy' on

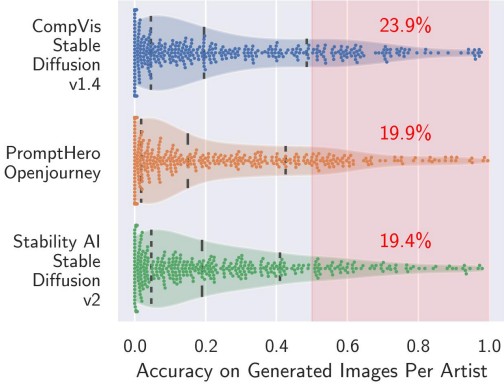

Figure 8: DeepMatch on generated art. In red: the fraction of artists with their styles recognized in at least half of their respective generated images.

generated images per artist, where accuracy is now interpreted as the rate which images generated to copy an artist are classified as that artist. In red, the fraction of artists who see accuracies of at least $50\%$ (i.e. so that the generated image *set* is classified to the original artist) are denoted per model, which we call the match rate. We observe an average match rate of $20.2\%$, indicating that for the vast majority of artists in our study, *simple prompting of generative models does not reproduce their styles* in a way recognizable to DeepMatch, which has an $89\%$ match rate on real art. For all three models, over half the artists see accuracies below $20\%$, with $26\%$ of artists seeing an average accuracy below $5\%$ for generated images. On the other hand, a handful of artists' styles are matched with high confidence: 16 artists see average accuracies over $75\%$. These include ultra famous artists like Van Gogh, Claude Monet, Renoir, which we'd expect generative models to do well in emulating. However, a few relatively lesser known artists are also present, like Jacek Yerka, who are still alive, and thus could be negatively affected by generative models reproducing their styles.

Matched Tag for Gustave Loiseau: landscape, simple colors, impressionistic
0 other artists also have this signature

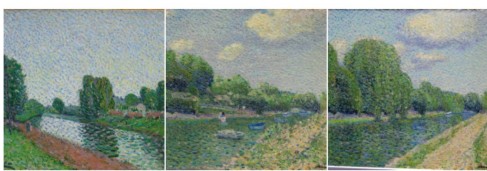 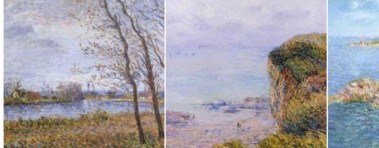

22 generated images with this signature      6 real images with this signature

Matched Tag for Salomon Van Ruysdael: boats, Dutch influences, smooth texture/application
0 other artists also have this signature

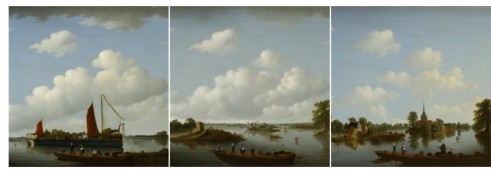 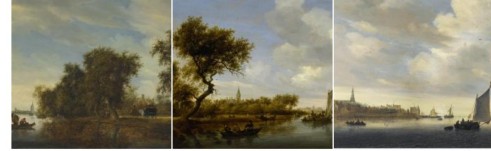

21 generated images with this signature      15 real images with this signature

Figure 9: Examples of applying TagMatch to generated images. TagMatch is inherently interpretable with respect to tags, as each inference comes with the exact set of tags that are (i) shared between the sets of test art and art from the predicted artist, and (ii) used to predict the artist.

In Appendix H, we design a small scale human evaluation to validate the outputs of our automated approach. We observe that generated art can fail to meet the threshold of style copying in two ways: either the generated art does not match the artist's style, or the style of the generated art is no more similar to the artist than to that of a highly similar artist. This underscores that style similarity alone does not constitute infringement, which instead requires replication of unique style signatures.

With **TagMatch**, in addition to predicting an artistic style, we can also articulate the specific tag signature shared between the test set of images and the reference set of images for the predicted style. Thus, we can inspect the shared signature, as well as instances from both sets where the signature is present, providing direct evidence of the potential style infringement a broader audience to independently verify. Inspecting some examples in figure 9 (more in fig. 16), we observe that while pixel level differences are common across retrieved image subsets, stylistic elements are consistent in both sets with the labeled tags, echoing our motivating claim that style copying goes beyond image or pixel-wise similarity. Lastly, TagMatch also allows for understanding image distributions from the perspective of interpretable tags. We explore this direction in appendix F.2, finding differences in the uniqueness of the tags present in generated art vs real art.

## 6 CONCLUSION

In our paper, we rethink the problem of copyright infringement in the context of artistic styles, which currently have no legal protection. So that our analysis is valuable to the relevant stakeholders, we base our method in legal precedent and take steps to ensure the interpretability of our approach. Then, we study the scientific basis for three questions: **i.** Do unique artistic styles exist? **ii.** Can unique styles be articulated interpretably in a quantitatively-grounded manner? **iii.** Do these unique styles re-appear in generated art? We answer these questions empirically, using a simple logical argument at the heart of our methods: if an artist's works are consistently recognized as their own, the artist has a unique style. Having reformulating the task to a classification problem over image sets, we develop `ArtSavant` – a novel tool to extract and detect artistic style *signatures* in a manner that is quantitative, intepretable, and rooted in legal principles. Crucially, we develop `ArtSavant` not to replace human decision makers, but instead to aide them in navigating the nuance associated with style copying. We find evidence of the existence of artistic styles, and in an empirical study, quantify the degree to which styles are potentially infringed, validating our framework. We hope our contributions help bridge the gap between legal and technical communities, towards quantitatively examining the nuanced issue of artistic style infringements.

## 7 ACKNOWLEDGMENTS

This project was supported in part by a grant from an NSF CAREER AWARD 1942230, ONR YIP award N00014-22-1-2271, ARO's Early Career Program Award 310902-00001, Army Grant No. W911NF2120076, the NSF award CCF2212458, NSF Award No. 2229885 (NSF Institute for Trustworthy AI in Law and Society, TRAILS), a MURI grant 14262683, an award from meta 314593-00001 and an award from Capital One.

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

## A    LIMITATIONS

Our work tackles a novel problem of artistic *style* infringements. Style, however, is qualitative. We merely put forward one definition for artistic style, along with two implementations for demonstrating the existence of a style given example works from an artist and recognizing the identified style in other works. Importantly, we argue that an artist's style is unique if we can consistently distinguish their work from that of other artists. However, we can only proxy the entire space of artists. We construct a dataset consisting of works from 372 artists spanning diverse schools of art and time periods in attempt to represent the space of existing artists, though of course we will always fall short in capturing all kinds of art. We provide tools to allow for this dataset to grow with time, and we caution that if only one artist for some broader artistic style is not present in our reference set, the uniqueness of that artist's style may be overestimated, and as such, generated images may be matched to this artist with an overestimated confidence. However, if only one out of 372 artists exhibits some style, than one could argue that that alone reflects a notable uniqueness of that artist. To employ a stricter criterion for alleging style copying, we'd recommend augmenting the reference set to include more artists with very similar styles to the artist in question. Nonetheless, we believe our reference dataset representing a large swath of existing art, to where analysis based on this reference set is still informative.

We also note that our atomic tagging leverages an existing foundation model (CLIP) with no additional training. While we verify the precision of our tags, CLIP is known to have issues with complex concepts. Further, we do not claim our tags achieve perfect recall (most image taggers do not). We advise users to interpret the assignment of a tag to indicate a strong presence of that concept, relative to similar concepts (i.e. from the same aspect of artistic style). While our tagger is not perfect, it is objective and automatic, enabling interpretable style articulation and detection. Also, we note that the field of image tagging in general has seen rapid improvement in the past year Huang et al. (2023), and an improved tagger could easily be swapped into our pipeline.

Lastly, we only analyze generated images using off-the-shelf text-to-image models. It is possible that particularly determined and AI-adept style thiefs fine-tune a model to more closely replicate specific artistic styles. This is a much more threatening scenario, though requires greater effort and ability by the style thief. We elect to demonstrate the feasability of our approach in the more broadly accessible setting of using models off-the-shelf, and note that our method can flexibly accept generated images produced in a different way (or perhaps discovered on the internet); notice generated images are an optional input in figure 7. We look forward to explorations of more threatening scenarios in future work, and hope both our formulation and methods for measuring style copying prove to be of use.

## B    A BRIEF HISTORY OF COPYRIGHT LAW: SITUATING OUR WORK

Throughout its history, copyright law has been centrally concerned with and affected by the replacement of slow, expensive, inexact human copying with comparatively fast, cheap, and exact machine copying. The very origin of copyright law is tied to the replacement of hand copying of manuscripts by scribes to machine duplication with printing presses. As the use of presses grew, publishers and authors became aware that printing enabled wider distribution of books. At the same time, however, they became aware that inexpensive unauthorized copying of those books would threaten the compensation available to an author for their initial investment in writing a book (Robinson, 1991; Loewenstein, 2010). The first copyright laws – the Statute of Anne in England in 1710 (sta, 1710), and the Copyright Act of 1790 (cop, 1790) in the United States – grew directly out of those concerns.

As machine copying has spread to other types of creative works, copyright law has adapted by including those works as copyrightable subject matter. In the mid-nineteenth century, chromolithography enabled dramatically better and less expensive copying of paintings. Congress responded in 1870 by extending copyright protection to paintings (cop, 1870; Brauneis, 2020). In the twentieth century, the development of audio and visual recording technologies presented new opportunities but also new threats to performers – musicians, actors, dancers, and other performing artists. Because recorded performances could in many cases be an adequate substitute for live performances, new markets opened, but both new and old markets would be threatened if recordings could be made and distributed without authorization. Congress responded by extending federal copyright protection to motion pictures in 1912 (Government, 1912), and to sound recordings in 1971 (Government, 1971). In the case of sound recordings, Congress also distinguished between human and machine copying.

Federal copyright law permits human copying of the sounds in a sound recording, through human imitation of those sounds that are independently recorded, while it prohibits direct electronic or mechanical machine copying of a sound recording (Government). That is another layer of recognition of the particular threat to authors posed by inexpensive, exact machine copying.

Generative AI is the newest technology for making creative works. Current generative AI tools are not designed to enable copying of specified preexisting works. However, for the first time, they may replace expensive human imitation of individual artistic styles with inexpensive machine mimicking of those styles. There have always been talented human artists who, with enough time for study and painstaking manual imitation, can produce believable forgeries – works that are not copies of any previously known works of well-known artists, but that convincingly apply the style of those artists to a different subject matter. In the twentieth century, artists such as Han van Meegeren (Lopez, 2008), Wolfgang and Helene Beltracchi (Birkenstock, 2014), and Shaun Greenhalgh (Greenhalgh, 2015) became infamous for their ability to create convincing forgeries of works of other artists. However, those forgers took weeks or months to create their forgeries, and they were motivated to make that considerable investment of time only because they believed they could sell the results for tens or hundreds of thousands of dollars.

By contrast, generative AI tools can create images, and texts and music in minutes or hours, at total machine and labor costs of only tens or hundreds of dollars, rather than tens or hundreds of thousands of dollars. Generative AI training algorithms are capable of modeling extraordinarily complicated patterns in a set of training works, and thus there is some reason to think that they could model the style of an author as manifested in a set of that author's works. There is also anecdotal evidence that generative AI tools can in fact produce new works that audiences mistakenly believe were created by particular human artists, authors, or performers (Giacomo, 2023; Coscarelli, 2023). Those capabilities raise the issue of whether copyright law should step in and protect copying of individual artistic styles, as it has stepped in many times in the past to protect copying of different types of individual works. That could be accomplished through legislation, but also possibly judicially. Because copyright infringement standards have never been codified, courts have created and modified those standards themselves. In the case of protection of fictional characters, courts have already gone beyond the individual work of authorship, and protected structures or entities that are only fully developed in multiple works (Kli, 2014; v. American Honda Motor Co., 1995; v. Towle, 2015).

Our purpose in this paper is not to take a position on the issue of whether copyright law should be extended to protect individual artistic styles, or on whether such an extension could be accomplished through legislation or through judicial decision. Neither is our purpose to automate the analysis of copyright infringement. Rather, we are interested in investigating whether there is any scientific support for the idea that there are identifiable individual artistic styles, and that those styles could be correlated with a group of human-understandable stylistic terms. Anecdotes about a generative AI tool producing an image or text that some people come to believe was created by a particular human artist or author are not proof that all or even some artists or authors have stable individual styles that can be modelled and applied in a wide variety of images, text, or music. However, when we try to train a model that can correctly identify authors of previously unseen works, we may get closer to understanding whether and when individual artistic styles exist. If we can link that classification to a selection of terms describing characteristics of those works, then we can explain to human beings what the components of such individual artistic styles might be. If some kind of protection of individual style ever became part of copyright law, judges or juries would still have to decide whether a particular output of a generative AI tool too closely mimicked the individual style of an artist. The degree to which an AI model could or could not correctly identify the author of a work, and the stylistic terms that that model could or could not correlate with that classification, would simply provide additional information to those human decision-makers.

## C   A NUANCE IN ARTISTIC STYLE INFRINGEMENTS: EXISTING ARTISTS CAN HAVE VERY SIMILAR STYLES

A crucial step in arguing that an artist's style has been infringed is to first demonstrate the existence of the given artist's *unique* style. We note that doing so objectively is non-trivial, as a style may not have a clear definition, and thus, it can be challenging to systematically compare to all other artistic styles, so to show uniqueness. In our work, we utilized classification, claiming that if an artist's works

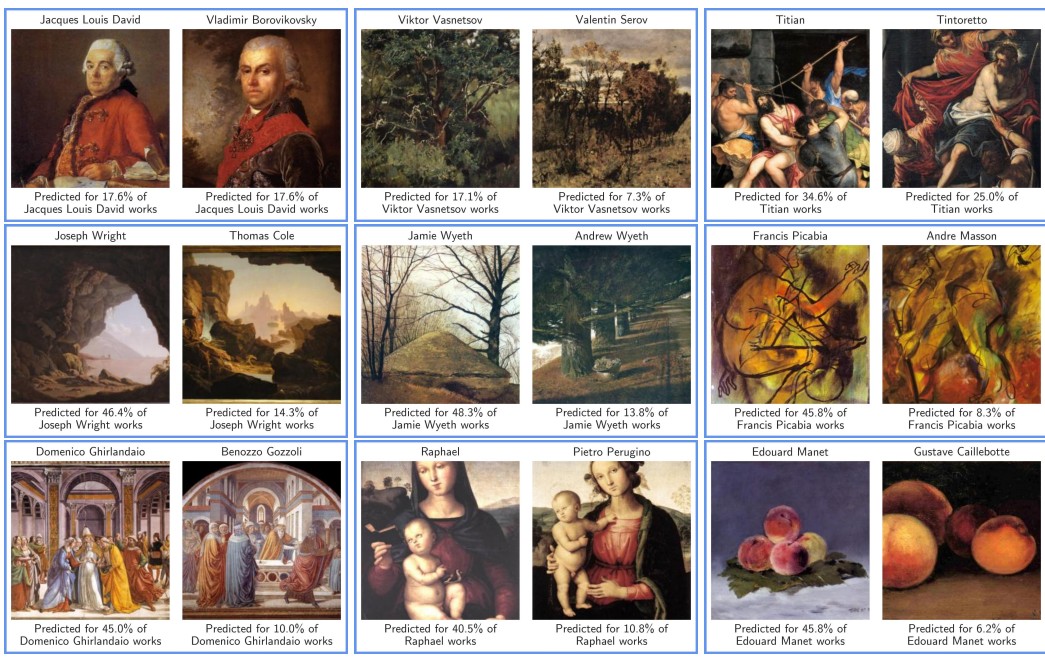

Figure 10: Examples of artists who's styles were not recognized by DeepMatch (i.e. less than half of their held-out works were predicted to the artist). Each panel shows an example work from (left) the unrecognized artist and (right) the artist that is incorrectly predicted most frequently over works from the unrecognized artist. We see that artists can use very similar, at times arguably indistinguishable, styles.

can consistently be mapped (i.e. at least half the time) to that artist (over a large set of other artists), than that artist must have some underlying unique style (parameterized by a neural signature).

In doing so, we found that $89.3\%$ of artists could be recognized based of a set of (at least 20 of) their works (held-out in training the classifier). What about the remaining $10.7\%$ of artists? We now take a closer look at these artists, and also introduce a second, stricter style copying criterion. Namely, we consider the notion that it may be unfair to claim a generative model is copying the style of an artist, if another existing artist seems to also be copying that artist. That is, we propose a way to verify that the generative model not only shows a substantial similarity to the copied artist, but also an *unprecedented* similarity.

## C.1 ARTISTS WHO'S STYLES WERE NOT RECOGNIZED

First, we inspect more examples from artists who were not recognized using our majority voting threshold in DeepMatch. That is, less than half of their held-out works were predicted to them. Figure 10 shows a number of examples, from which we can make some qualitative observations. First, the styles of artists who operate in the same broader genre (e.g. portraiture, landscapes, narrative scenes in renaissance styles, etc) can be extremely similar. We even see an instance where an artist's son's style is indistinguishable from his father's (Jamie and Andrew Wyeth). Lastly, we note that in most cases, the artists only marginally fall short of our recognition threshold (i.e. accuracy for their held-out works is only a bit below $50\%$). We utilize majority voting because (i) it is intuitive, (ii) it requires *consistent* appearance of the neural signature across works, and (iii) it allows for abstention when no particular style is strongly present. However, the exact threshold of $50\%$ can be altered as desired. In summary, as in Figure 4, we see artistic styles can be very similar, making the existence of unique artistic styles for the vast majority of artists a non-trivial observation.

If an artist's style cannot be recognized over their own held-out works, arguing that a generative model copies that style is strenuous, as the style itself is ill-defined. Notably, in these cases, the classifier had an option to predict the correct artist. However, in applying DeepMatch to generated images, there is no direct option for the classifier to abstain from predicting anyone, under that generated

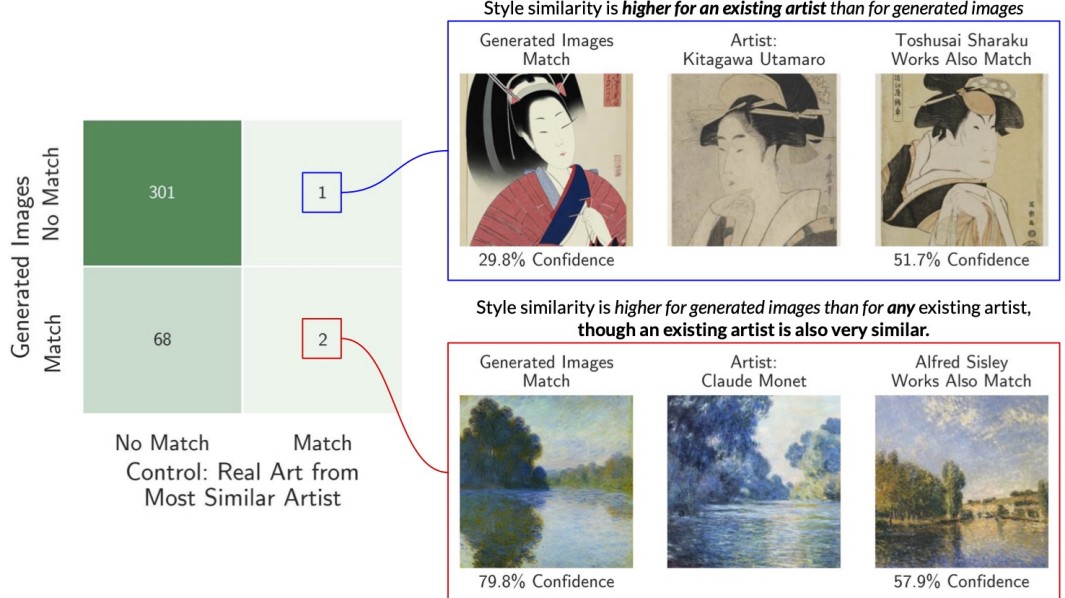

Figure 11: We verify the stricter criterion of *unprecedented similarity* by holding out the real artist with highest similarity to a given artist, and checking if the held-out real artist's works are flagged as potential style copying by DeepMatch. (**left**) We observe only three artists where the most similar held-out artist has their work flagged as a style match, and in all cases, when generated images are flagged, the match confidence of the generated images exceeds that of the held-out real artist's works (i.e., **the generated images flagged by our method reflect *unprecedented similarity* to the given artist's style**). (**right**) Inspecting the flagged held-out artists further show that style copying is very nuanced, as artists take inspiration from one another, and as such, they may already have very similar styles. While we always observe unprecedented similarity, a potential solution to style copying may be for generative models to ensure that they do not copy any more than what already exists; that is, they may exhibit some copying, but no more than for which precedent already exists.

art comes from a "new artist", which takes inspiration from existing artists. Note that abstention is still possible (due to the majority voting in DeepMatch), and occurs when a match confidence falls below $50\%$. To make comparisons fairer to generative models, we now discuss a stricter criterion of *unprecedented similarity*.

C.2 *Unprecedented Similarity*: Do generative models copy styles more than existing artists already do?

A nuance that requires consideration when studying artistic style copying is that it is possible for two artists to have very similar styles. Thus, it may be unfair to allege that a generative model is copying an artist $a$ if there exists another artist $b$ who's style is just as or in fact even more similar to artist $a$. Towards this end, we introduce *unprecedented similarity*, which requires that the similarity between works of a generative model $A'$ and works of the artist inteded to be copied $A$ is higher than the similarity of any existing artist with $A$. That is, $sim(A, A') \geq sim(A, B)$ for works $B$ from all other existing artists $b$.

Note that this is a stricter criterion than our previous threshold. In DeepMatch, we required that at least half of the works in a given set of test images were predicted to a single artist in order for us to flag the test images as a potential style infringmenet. In other words, that threshold required that $sim(A, A') \geq 0.5$, which in turn implies that $sim(A, A') \geq sim(A', B)$ for all $B$ (with room to spare; here we use match confidence to denote similarity).

Now, however, instead of just comparing $A'$ to all $B$, we must also compare all $B$ to $A$. Instead of comparing all other artists, we inspect the most similar artist $b^*$ to $a$, identified by taking the artist $b$ with the highest rate of false positive predictions to artist $a$. Then, we hold out $b$, and train a new

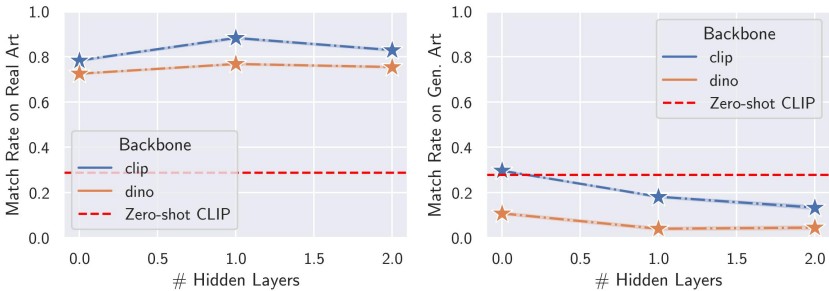

Figure 12: Alternate implementations of DeepMatch, using DINOv2 and CLIP backbones, and varying the number of hidden layers. We also present performance of zero-shot CLIP. Numbers are averaged over five trials, except for zero-shot CLIP, which is deterministic.

classifier on the remaining 371 artists. Finally, we check for style matches of for the set of generated images $A'$ and the works $B^*$ from the most similar artist $b^*$.

Figure 11 summarizes our result for OpenJourney (all three models studied show consistent results). We find that only in three cases do we see a held-out artist's work flagged as potential style copying. Notably, in all instances where generated work is flagged as potential style copying, the corresponding held-out artist's work is either not flagged or is flagged with lower confidence, indicating that the instances of style copying of generative models that we observe always also satisfy the criterion of unprecedented similarity.

Taking a closer look at instances where held-out art is flagged for style copying (or perhaps style emulation?), we again see just how similar the works of different artists can be. Namely, we see that some artists works seem to fall into a broader genre of art that many artists utilize (e.g. ukiyo-e or impressionism). In summary, while generative models can very closely resemble the style of a given artist, contextualizing copying by generative models with respect to copying (or perhaps, 'style emulation') already done by existing artists is crucial in order to afford the same artistic liberties to generative models as have been provided to other artists in the past.

## D    BASELINES

We now present some alternate implementations to the methods we present, so to serve as baselines. We note that a key contribution of our work is reformulating the problem of detecting style infringements from computing image-wise similarity to performing classification over image sets, and building a tool around this idea. Thus, it is rather challenging to perform apples-to-apples comparisons to prior copy detection works, as our methods implement a different task. We include substantial qualitative discussion comparing our approach to image-similarity techniques (and thus motivating our framework) in section 3, and we add to that discussion here.

We further stress that there is not a singular numerical objective that we can use as a way to compare methods. For example, we report the accuracy of matching artists (i.e. aggregating classification predictions with majority voting), but since it is not necessarily true that all artists are distinguishable, it would be imprudent to strictly prefer a higher accuracy, as there is no strict groundtruth; that is, there is no completely definitive way to say if an artist has a unique style or not, due to the subjective/qualitative nature of style. Nonetheless, for lack of other quantitative metrics, we inspect accuracy on real and generated images for a few lightweight approaches to artist classifications, and compare them below.

### D.1    DEEPMATCH

Figure 12 shows the performance of different classifiers, where we vary the frozen backbone and the number of hidden layers. We find that classifiers trained on CLIP yield higher match-rates for both real and generated art than classifiers train on DINOv2 Oquab et al. (2024) embeddings. Interestingly, zero-shot CLIP does poorly on real art, but well on generated art, perhaps because many

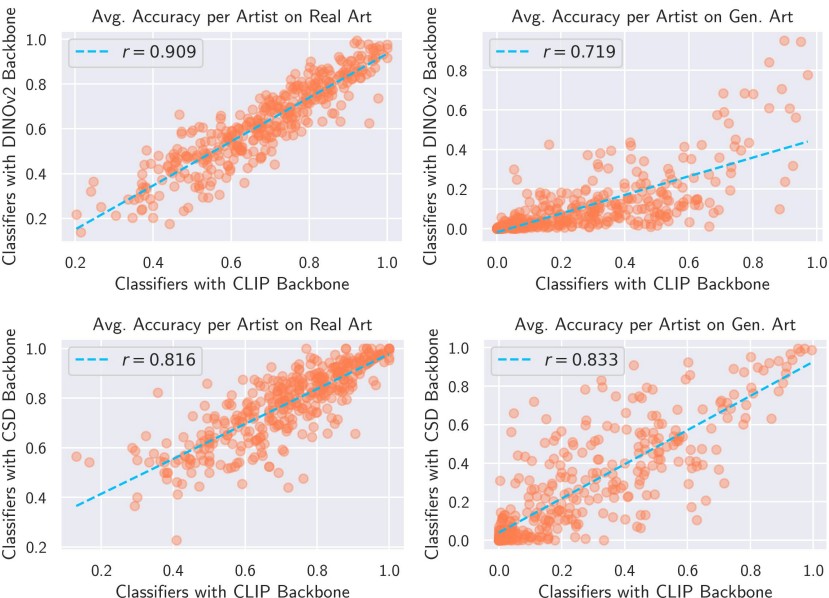

Figure 13: Per-artist accuracy for classifiers using CLIP and DINO backbones (**top**) or CLIP and CSD Somepalli et al. (2022) (**bottom**) are highly correlated. While each classifier may yield different overall accuracy, the *relative* notions of (i) how recognizable the artist's real art is, and (ii) how much so the artist's style appears in generated works, appear to be classifier agnostic.

generative models optimize using CLIP-score, which applies the same mechanism as zero-shot CLIP classification, perhaps explaining the assertion that generative models are highly capable of imitating humans found in this brief work Casper et al. (2023). The number of hidden layers does not have a very strong affect on recognizing real art, but it does appear inversely related to the ability of the model to recognize generated art. It is possible that having two many hidden layers can overfit the model to the distribution of real images, creating a distribution shift when applied on the generated images.

While exact numbers seem to vary, we note that relative trends (i.e. between artists) appear agostic to the underlying classifier. Figure 13 shows accuracy per artist for classifiers trained on CLIP vs DINOv2 embeddings, as well as classifiers trained on CLIP vs. CSD (a style fine-tuned version of CLIP) embeddings (Somepalli et al., 2022). For both real and generated art, the per-artist accuracies are strongly correlated, which could motivate using relative metrics in addition to absolute values dependent on exact accuracy values; note that we include relative numbers in our `ArtSavant` report (see Figure 1; e.g., 'percentile of recognizability').

We ultimately choose something in the middle of the road: a 1-hidden layer MLP on CLIP embeddings, which has the strongest performance recognizing real art, and appears to have some ability to recognize generated art. We note the majority aggregation that we apply is just one way to summarize the classification output across an image set. We opt for it because it is intuitive and it provides a natural avenue for abstention, though this threshold can be modified as desired, and inspecting relative accuracies could be most informative. We again stress that our current implementation serves as a proof of concept of our framework, which is our primary contribution.

## D.2 TAGMATCH

We now present baselines for TagMatch. Like above, and indeed more so, accuracy is not exactly an objective to maximize. In fact, what is most important with TagMatch is interpretability, and ease with which the output of TagMatch can be used in arguments to a broader, non-technical audience. Thus, we consider a popular framework from the interpretable classification literature: concept bottleneck models (CBM) Koh et al. (2020). Namely, we train a linear layer atop concept predictions extracted from CLIP, so to create a CBM without direct concept supervision, as in Moayeri et al. (2023); Yuksekgonul et al. (2023).

|  | CBM | CBM + sparsity | Ours |
|---|---|---|---|
| Accuracy on real art | 62.8% | 58.7% | 61.5% |

Table 1: Baselines for TagMatch

As shown in table 1, accuracies are roughly similar. But, the interpretability provided by the methods are markedly different. CBM involves examining the final linear layer to discern which concepts are important to which class, which requires users to inspect a coefficient for every concept. Adding sparsity by way of an $\ell_1$ penalty can help, but the problem persists. Our version of TagMatch, on the other hand, affords concise articulations of tag signatures, as well as a number of how many other artists share a given signature. Perhaps most crucially, our implementation also yields faithful attribution, which can be critical in gathering evidence to present to a judge or jury.

### D.3 EVALUATION ON IMAGE RETRIEVAL

We now present a comparison between our method, DeepMatch, and baseline methods such as CLIP zero-shot and CSD (a version of CLIP fine-tuned to better measure style similarity (Somepalli et al., 2024)). We evaluate all methods on how well they can retrieve images from a fixed reference set of artworks based on their similarity with a query artwork. We draw the query artworks from the test split of our dataset, and the fixed reference set is the training split. We count each retrieved artwork created by the same artist as the query artwork as a true positive, and if created by a different artist as a false positive. This setup mirrors one potential usage of these tools in the real world, wherein if the AI-generated artwork contains some distinctive artistic signature belonging to a real artist, we would expect the tool to retrieve artworks which also contain the same artistic signature. As DeepMatch is not originally intended for retrieval, we repurpose it's classifier for retrieval purposes by using the softmax probabilities it outputs given an image as an encoding of that image (i.e. each image is represented by a 372-dimensional vector, where the $i^{th}$ element corresponds to the likelihood that the image is authored by artist i).

In Figure 14, we plot the distribution of AUROCs over the artists in our dataset for the baseline methods and DeepMatch. We find that CSD (mean AUROC of $0.891$) outperforms CLIP (mean AUROC of $0.861$) in this task, which is expected as CSD outputs a more style-centric similarity measure compared to CLIP. However, CSD is not specifically aware of what makes an artist's style distinctive and thus struggles to retrieve the most relevant artworks compared to DeepMatch (mean AUROC $0.979$), which performs the best among the three.

Intuitively, the improved performance of our method can be linked to the original objective of the three methods considered. Image similarity methods (such as using cosine similarity between CLIP embeddings as similarity score) are not equipped or intended to measure stylistic similarity, but rather only a general sense of image similarity. Even methods such as CSD which are specifically trained using a contrastive loss to be invariant to style preserving transformations are not aware of the components that constitute a *unique* artistic style. That is, while CSD is trained to contrast general art styles (e.g. "impressionism" from "cubism"), it is not trained to contrast between two (potentially very similar) artists (e.g. "*Manet*'s impressionism" vs. "*Monet*'s impressionism"). From a copyright perspective, we are precisely interested in only those stylistic components that set an artist's work apart from their peers - that is, a unique artistic signature. This naturally suggests a framework which analyzes style from the lens of image classification. Our method, DeepMatch, is trained to classify artworks and thus upweights stylistic features which are most unique and thus useful for classification.

We note another factor that may also contribute to our improved performance is that our method is specifically trained for distinguishing the 372 artists in our datasets, while our baselines are more general. However, we engineer our method to be very efficient, allowing for new artists to be introduced as needed, at minimal cost. Again, this was precisely be design, as in practice, we envision only one or a few new artists would need to be incorporated given a copyright case to be studied. Overall though, the key distinction between our method and image-similarity based techniques is that (1) we emphasize capturing features that distinguish between individual styles (as opposed to broader

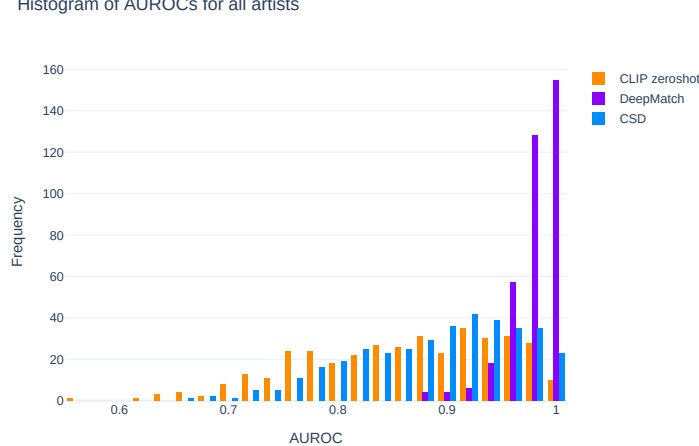

Figure 14: Distribution (over artists) of AUROCs for the task of retrieving art from the same artist as a query image, using DeepMatch, CSD, and CLIP zero-shot to encode images.

ones), and (2) we account for the fact that style is comprised over a body of work, going beyond comparing just a pair of images together.

### D.4 STABILITY

We also explore the stability of our method to using different data splits. We perform five different random train / test splits, and inspect the accuracy of our implementations of DeepMatch and TagMatch. DeepMatch per-image accuracies are very stable, with a standard deviation of $0.1\%$. TagMatch is also stable, though less so, with a standard deviation $1.1\%$.

## E ON ALTERNATE PROMPTS

We briefly explore using alternate prompts to generate images. Namely, we create 120 prompts of the form "{an object} in {location} in the style of {artist}" (e.g. "A bottle in forest in the style of Jeff Koons", which are by nature no longer artist-specific (like the titles we originally use). Using DeepMatch, average match rate drops considerably in this less specific case, from $20\%$ to $8\%$. This is in line with existing wisdom that prompting can significantly affect the behavior of a model, and also echoes our overall empirical observation that current style copying does not appear to be very prevalent. We hope that our framework can be useful in examining which prompts induce greatest copying going forward, especially as prompt and model sophistication grows.

## F DETAILS ON TAGMATCH

We now provide greater details regarding the implementation of TagMatch, a central technical contribution of our work. TagMatch is a method to classify a set of images to a class; specifically, we map a set of artworks to one artist, selected over 372 choices. TagMatch is not as accurate as DeepMatch, as it maps held-out works of each artist in our WikiArt dataset to the correct artist about $61\%$ of the time (compared to $89\%$ top-1 accuracy for DeepMatch). However, top-5 accuracy is more reasonabe, achieving above $80\%$. Most notably, **TagMatch is inherently interpretable and attributable**. It consists of three steps: (i) assigning atomic tags to images, (ii) efficiently composing tags to obtain more unique *tag signatures*, and (iii) matching a test set of images to a reference artist based on the uniqueness of the tags shared between the test set and works from the predicted reference artist.

Our method is fast and flexible: after caching image embeddings, the whole thing only takes minutes, and it is easy to modify the concept vocabulary as desired, as the tagging is done in a zero-shot manner. Through MTurk studies, we verify that the atomic tags we assign our mostly precise, though we recognize that these descriptors can be subjective. Thus, while we do not claim perfect tagging, we stress that our method is easy to understand, and crucially, is deterministic per image. Therefore, ideally our tagging may be more reliable biased than human judgements, particularly when the humans involved may be biased (e.g. an artist alleging copying and a lawyer defending a generative model would have strong and opposing stakes).

Below, we provide details for image tagging (§F.1), artist tagging (§F.2), artistic style inference via tag matching (§F.3), effect of hyperparameters (§F.4), details on efficiency (§F.5), and a review of validation (§F.6).

## F.1 IMAGE TAGGING

As explained in §4.2, we utilize CLIP to attain a diverse set of atomic tags per image in a zero-shot manner. Specifically, we first define a vocabulary of descriptors along various aspects of artistic style. Then, given an image, we do selective multi-label zero-shot classification *for each aspect*. Performing zero-shot classification per aspect proves to be critical in order to achieve a diversity of tags and a similar number of tags per image. We find that some descriptors always lead to higher CLIP similarities than others. Specifically, descriptors for simple aspects, like colors and shapes, yield higher similarities than more complex aspects like brushwork and style. Thus, using a global threshold across descriptors would lead to a less diverse descriptor set. Moreover, we observe some images have higher similarities across the board than others, which again would lead global thresholding to result in a disparate number of tags per image. Our per-aspect scheme requires that the descriptors within each aspect are mostly mutually exclusive; we prioritize this in the construction of the concept vocabulary, via the prompt we present the LLM assistants and our manual verification.

Namely, we prompt both Vicuna-33b and ChatGPT with "*I want to build a vocabulary of tags to be able to describe art. First, consider different aspects of art, and then for each aspect, list about 20 distinct descriptors that could describe that aspect of art. Please return your answer in the form of a python dictionary.* ". We then perform a filtering step with a human in the loop, where we manually remove tags that are difficult to recognize or redundant. After this filtering step, we add in a few new aspects. First, we incorporate the 20 *styles* (e.g., "impressionism") and *genres* (e.g., "portrait") that are most common amongst works in our WikiArt dataset; note that all WikiArt images also contain metadata for these categories. Finally, we add some easy to understand tags such as *color* and *shape* which can be important characteristics describing a given painting. The concept vocabulary we use is contains shown below:

- **Style**, caption template: *{} style*. Descriptors:
  - *realism, impressionism, romanticism, expressionism, post impressionism, art nouveau modern, baroque, symbolism, surrealism, neoclassicism, naïve art primitivism, northern renaissance, rococo, cubism, ukiyo e, abstract expressionism, mannerism late renaissance, high renaissance, magic realism, neo impressionism*
- **Genre**, caption template: *the genre of {}*. Descriptors:
  - *portrait, landscape, genre painting, religious painting, cityscape, sketch and study, illustration, abstract art, figurative, nude painting, design, still life, symbolic painting, marina, mythological painting, flower painting, self portrait, animal painting, photo, history painting, digital art*
- **Colors**, caption template: *{} colors*. Descriptors:
  - *pale red, pale blue, pale green, pale brown, pale yellow, pale purple, pale gray, black and white, dark red, dark blue, dark green, dark brown, dark yellow, dark purple, dark gray*
- **Shapes**, caption template: *{}*. Descriptors:
  - *circles, squares, straight lines, rectangles, triangles, curves, sharp angles, curved angles, cubes, spheres, cylinders, diagonal lines, spirals, swirling lines, radial symmetry, grid patterns*

- **Common Objects**, caption template: *{}*. Descriptors:
    - *male figures, female figures, children, farm animals, pet animals, wild animals, geometric shapes, fruit, vegetables, intsruments, flowers, boats, waves, roads, household items, the moon, the sun, saints, angels, demons*

- **Backgrounds**, caption template: *{} in the background*. Descriptors:
    - *fields, blue sky, night sky, sunset or sunrise, forest, rolling hills, simple colors, beach, port, river, starry night, clouds, shadows, living room, bedroom, trees, buildings, chapels, heaven, hell, houses, streets*

- **Color Palette**, caption template: *{} color palette*. Descriptors:
    - *vibrant, muted, monochromatic, complementary, pastel, bright, dull, earthy, bold, subdued, rich, simple, complex, varying, minimal, contrasting*

- **Medium**, caption template: *the medium of {}*. Descriptors:
    - *oil painting, watercolor, acrylic, ink, pencil, charcoal, etching, screen printing, relief, intaglio, collage, montage, photography, sculpture, ceramics, glass*

- **Cultural Influence**, caption template: *{} influences*. Descriptors:
    - *Indigenous, European, American, East Asian, Indian, Middle Eastern, Hispanic, Aztec, Contemporary, Greek, Roman, Byzantine, Russian, African, Egyptian, Tahitian, Polynesian, Dutch*

- **Texture**, caption template: *{} texture*. Descriptors:
    - *rough, smooth, bumpy, glossy, matte, roughened, polished, textured, smoothed, brushstroked, layered, scraped, glazed, streaked, blended, uneven, smudged*

- **Other Elements**, caption template: *{}*. Descriptors:
    - *stippled brushwork, chiaroscuro lighting, pointillist brushwork, multimedia composition, impasto technique, repetitive, pop culture references, written words, chinese characters, japanese characters*

Now, we detail the implementation of our modified zero-shot classification. Recall that in zero-shot classification, one computes a text embedding per class, which amounts to the classification head, and computes an image embedding for the test input, so that the prediction is the class who's text embedding has the highest cosine similarity to the test image embedding. In computing the text embeddings, we take each descriptor (e.g. *Dutch*) and place it an aspect-specific caption template (e.g. *Dutch → Dutch influences*), and then average embedddings over multiple prompts (e.g. "artwork containing *Dutch influences*", "a piece of art with *Dutch influences*", etc), as done in Radford et al. (2021). We modify standard zero-shot classification to allow for the fact that more than one descriptor (or perhaps none) from a given aspect may be present. Namely, instead of assigning the most similar descriptor per-aspect, we assign an atomic tag for any descriptor who's similarity is significantly higher than other descriptors for that aspect. We achieve this via z-score thresholding: per-aspect, we convert similarities to z-scores by subtracting away the mean and dividing by the standard deviation, and then admit atomic tags who's z-score is at least $1.5$.

The template prompts we utilize for embedding each concept caption are as follows:

- art with
- a painting with
- an image of art with
- artwork containing
- a piece of art with
- artwork that has
- a work of art with
- famous art that has
- a cropped image of art with

---

**Algorithm 1** Iterative Algorithm to Obtain Tag Composition Per Artist $a \in \mathcal{A}$

---

**Require:** $\mathcal{D}_a$ (Images for artist $a$), $\mathcal{C}_a$ (Common tags for artist $a$)

   $\mathcal{S}_a = \{\}$              ▷ Stores the tag compositions with their associated counts

   **for** $x \in \mathcal{D}_a$ **do**

      $I(x) = \mathrm{tag}(x) \cap \mathcal{C}_a$         ▷ Compute the intersection with common atomic tags

      $\mathcal{P}(I(x)) = \mathrm{ComputePowerSet}(I(x))$       ▷ Compute power-set of the tags

      $\mathrm{UpdateCount}(\mathcal{S}_a, \mathcal{P}(I(x)))$       ▷ Update the count of each tag composition

   **end for**

   $\mathrm{Filter}(\mathcal{S}_a)$       ▷ Keep tag compositions which occur above a count threshold of 3

---

## F.2   FROM IMAGE TAGS TO *unique* ARTIST TAGS

Recall that we define styles not per-image, but over a set of images. Namely, we seek to surface tags that occur frequently. The best way to do so is to simply count the occurrences of each tag, and discard the ones that rarely appear. However, each atomic tag is not particularly unique with respect to artists. We utilized *efficient composition* of atomic tags to arrive at more unique tag signatures, as shown in figure 6 and detailed in algorithm 1. Importantly, we utilize a threshold here to differentiate what a common tag is; we require a tag to appear in at least three works for an artist in order for the tag to count as a frequently used tag by the artist. We note that tag composition can be done efficiently because we have a relatively low number of tags per image: on average, there are $6.2$ atomic tags per image. Moreover, because the number of occurrences for a composed tag is bound belo by the number of occurrences of each atomic tag in the composition, we can ignore all non-frequent atomic tags. Thus, we can iterate over the powerset of common atomic tags per image without it taking exorbitantly long. We include one fail safe, which is that in the rare instance where an image has a very high number of common atomic tags, we truncate the tag list to include only 25 tags. Over the $91k$ images that we encounter, this happens only once. We highlight that our tag composition takes inspiration from Rezaei et al. (2023).

## F.3   PREDICTING ARTISTIC STYLES BASED ON MATCHED TAGS

Once we have converted tags per image to tags per artist, we can then utilize these artist tags to perform inference over a set of images. Namely, given a test set of images, we extract common tags (including tag compositions) for the test set and compare them to tags extracted for each artist in our reference corpus. Then, we predict the reference artist who shares the most unique tags with the test set.

Figure 15 best explains our method, as it shows the documented code. We note that all code will be released upon acceptance. We'll now explain it step by step. First, for each artist and for the test set of images, we find common tags via (i) assigning atomic tags to each image, (ii) finding the commonly occurring atomic tags, (iii) counting compositions of the commonly occurring atomic tags, and (iv) discarding tags (including compositions) that do not occur frequently enough. The code shows this done for the test set of images; we perform this per reference artist when the `TagMatcher` object (for which `tag_match` is function) is initialized; notice fields like `self.ref_tags_w_counts_by_artist`, which contain useful information about the reference artists, computed once and re-used for each inference.

Then, we loop through the set of 'matched' tags (i.e. those that occur for both the test set of images and at least one reference artist), starting with the most unique ones. Here, uniqueness refers to the number of reference artists that frequently use a tag. For each tag, we loop through all artists that also use that tag. For the first $k$ (denoted by `self.matches_per_artist_to_consider` in the code) matched tags per artist, we add a score to a list of scores for the artist, which ultimately are averaged. The score contains an integer and a decimal component. The integer component is the number of reference artists that share the matched tag. The decimal component is the absolute value of the difference in frequency with which the tag appears, over the reference artist's works and the test set of images; note that this is always less than one. This way, when comparing two matched tags, a lower score is assigned to a more unique one, and one there is a tie in uniqueness, we break the tie based on how similar the frequency of the matched tag is for the test artist and reference artist.

```python
def tag_match(self, test_img_paths: List[str], test_artist: str):
    dset = BasicDsetFromImgPaths(test_img_paths, self.vlm.transform, dsetname=test_artist)

    tags_by_path = self.tag_images(dset)
    common_tags = self.find_common_tags(tags_by_path)
    composed_tags_w_counts = self.compose_tags(common_tags, tags_by_path)

    # Now we cross-reference the found tags w/ tags for reference artist
    counts_over_ref_artists_by_tag = dict({
            t:len(self.ref_artists_by_tag[t])
            for t in composed_tags_w_counts if t in self.ref_artists_by_tag
    })
    # We sort the tags by uniqueness: we first inspect tags that occur for the lowest number of reference artists
    counts_over_ref_artists_by_tag = dict(sorted(counts_over_ref_artists_by_tag.items(), key=lambda x:x[1]))

    # We will return a score per artist to resemble the typical output of a classifier
    scores_by_artist = dict({artist: [] for artist in self.ref_dset.artists})
    # We will also keep track of the tags used in computing the score per artist -- this provides faithful interpretations
    matched_tags_by_artist = dict({artist: [] for artist in self.ref_dset.artists})
    # Now we loop through each tag that also occurs for reference artists
    for t, num_ref_artists_w_tag in counts_over_ref_artists_by_tag.items():
        # For each tag, we loop through all matches (i.e. any reference artist that also has the tag)
        for ref_artist in self.ref_artists_by_tag[t]:
            # We only consider the top k most unique matched tags per artist (k = self.matches_per_artist_to_consider)
            if len(scores_by_artist[ref_artist]) < self.matches_per_artist_to_consider:
                # Compute frequency of matched tag over works from the reference artist
                num_works_of_ref_artist_w_tag = self.ref_tags_w_counts_by_artist[ref_artist][t]
                freq_for_ref_artist = num_works_of_ref_artist_w_tag / self.num_works_by_ref_artist[ref_artist]
                # Compute frequency of matched tag over works from the test artist
                freq_for_test_artist = composed_tags_w_counts[t] / len(tags_by_path)
                # Our score is the uniqueness of the matched tag + |diff in frequencies of tag for ref artist and test artist|
                scores_by_artist[ref_artist].append(num_ref_artists_w_tag + np.abs(freq_for_ref_artist - freq_for_test_artist))
                matched_tags_by_artist[ref_artist].append(t)

    # We set the score to inf for any artists that did not have enough matched tags
    scores = np.array([np.mean(scores_by_artist[artist][:self.matches_per_artist_to_consider])
            if len(scores_by_artist[artist]) >= self.matches_per_artist_to_consider else np.inf for artist in self.ref_dset.artists])

    # Finally, we return scores along with explanations for each artist
    return scores, matched_tags_by_artist
```

Figure 15: Code for predicting artistic styles via matched tags.

Finally, we average the list of scores per artist to get a single score per reference artist, analogous to a logit. We assign a score of `inf` for any artist with less than `self.matches_per_artist_to_consider` (which we set to 10) matched tags. This hyperparameter makes our tag matching less sensitive to individual matched tags, and empirically results in a substantial improvement in top-1 accuracy on held-out art from WikiArt artists (see next section).

### F.4 CHOOSING HYPERPARAMETERS

Overall, there are three hyperparameters to our method: the z-score threshold, the tag count threshold, and the number of matches to consider per artist. Here is quick refresher on what they each do:

- The z-score threshold determines how much more similar a descriptor needs to be to an image compared to other descriptors for the same aspect in order for the descriptor to be assigned as an atomic tag of the image. The value we use is $1.75$.

- The tag count threshold is the minimum number of an artist's works that a tag needs to be present in order for a the tag to be deemed common for the artist. The value we use is $3$.

- The number of matches to consider per artist pertains to how many matched tags are considered when computing the final score per artist in tag match. That is, the final score for an artist is the average of the top-k most unique tags that the artist shares with the test set of images, where $k$ corresponds to this hyperparameter. The value we use is $10$.

Now that the role of each hyperparameter is clear, let's discuss how hyperparameters can be adjusted towards particular ends, along with the potential consequence of each action:

- To increase the number of atomic tags, lower the z-score threshold. Risk: atomic tags may be less precise, and the method will take longer to run, as there will atomic tags and composed tags.

Matched Tag for Antoine Blanchard: simple colors, streets, Contemporary influences, social symbolism
0 other artists also have this signature

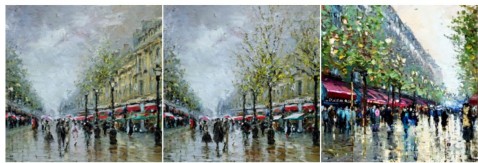 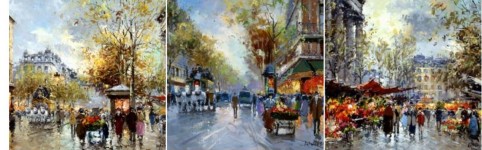

8 generated images with this signature      5 real images with this signature

Matched Tag for Franz Xaver Winterhalter: broad brushwork, female figures, historical symbolism
0 other artists also have this signature

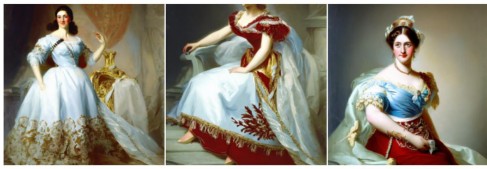 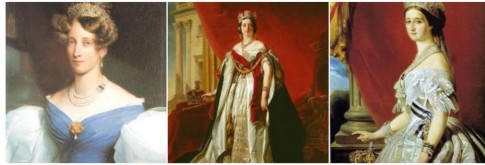

19 generated images with this signature      7 real images with this signature

Matched Tag for Arthur Rackham: illustration, children, fantastical subject matter
0 other artists also have this signature

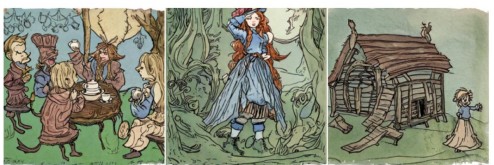 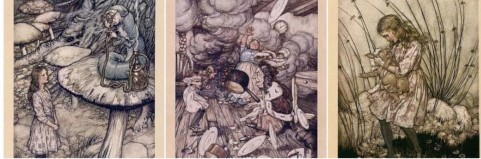

12 generated images with this signature      5 real images with this signature

Matched Tag for Nicholas Roerich: geometric shapes, simple colors, geographical symbolism
0 other artists also have this signature

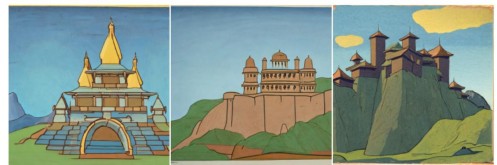 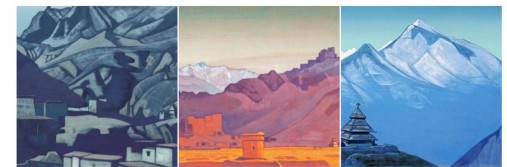

47 generated images with this signature      22 real images with this signature

Figure 16: Additional examples of applying TagMatch to generated images.

- To get more tags per artist, lower the tag count threshold. Risk: some tags will become less unique. Other tags will be introduced, and may be very unique, which could skew tag matching. Also, the method may take longer to run, as there will be more tags.

- To make inference less sensitive to a low number of matched tags, increase the number of matches to consider per artist. Risk: when you consider more matches, interpretation is a little more difficult, as you have more reasons for each inference, and it will take longer to view them all.

To choose hyperparameters, we selected a small range of reasonable values and swept each hyperparameter individually. While a combined search would likely yield better accuracy numbers, we opt out of hyper-tuning TagMatch for accuracy, as its main objective is to provide and interpretable and attributable complement to DeepMatch. We find the (relatively strong, considering the high number of artists considered) accuracy numbers encouraging, but do not find it a priority, as DeepMatch arguably provides a stronger and easier to understand signal of *if* style copying is happening. TagMatch, on the other hand, tells us *how* and *where* it is happening (if observed with DeepMatch).

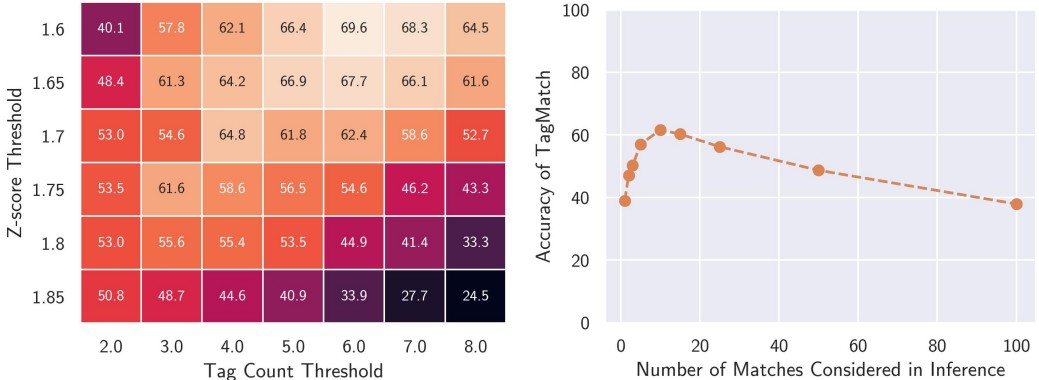

Figure 17: Sweep of hyperparameters asssociated with TagMatch. **(left)** We jointly sweep the z-score threshold and the tag count threshold. **(right)** Having fixed the first two parameters, we sweep the last one: the number of matches considered in inference. See detailed discussion in §F.4.

We also include a hyperparameter sweep, of the z-score threshold and tag count threshold jointly, and of the number of matches to consider separatedly afterwards. Figure 17 visualizes the results. Choosing a lower z-score threshold results in higher TagMatch accuracies. However, a lower z-score threshold would admit a greater number of false positive tags, and also incurs a longer time of computation, as there are more tags to compose (we empirically observe an increase of about 50% in run time using our 372 artist reference corpus). Increasing the tag count threshold can reduce the time of computation and also increase sensitivity to false positive tags (on individual images), resulting in higher TagMatch accuracies. Interestingly, considering more matches improves accuracy considerably, but eventually saturates and reduces accuracy. Essentially, by considering more matches per artist, inference becomes less sensitive to the most unique matched tag between the artist and the test set. The smoothed predictions are more accurate up to a point (i.e. 10 matches), but then hinder accuracy. Also, choosing too high a number here can make faithful interpretation more cumbersome, as there are more matches to inspect afterwards.

We reiterate that the main goal of TagMatch is not to be super accurate, but to complement DeepMatch with interpretations (via matched tag signatures) and attributions (via works from the test set and from the reference artist that present the matched tags). We ultimately first choose a high z-score threshold of 1.75, as a preliminary check revealed this threshold to have considerably higher precision in its atomic tags (which we validate with a human study), and since it speeds up the analysis. Then, we choose the best tag count threshold (3) and number of matches to consider (10), in that order. We hope our discussion of the impact of each hyperparameter can enable practitioners to modify these choices as they please. Furthermore, as base VLMs and tagging methods improve, our framework can modularly swap out our zero-shot tagging (and thus also the z-score threshold) for a stronger method, while retaining the other structure of TagMatch.

### F.5    EFFICIENCY OF TAGMATCH: RUNS IN ROUGHLY 1 MINUTE

TagMatch is surprisingly fast. The longest step by far is computing CLIP embeddings for the reference artworks. This takes us about 5 minutes using one rtx2080 GPU with four CPU cores to embed the $73k$ training split images using a CLIP ViT-B\16 model. Importantly, this step is done only once, and in practice, is done offline. The other steps and approximate time needed for each are as follows: embedding concepts (5 seconds), extracting common atomic tags and composing them (45 seconds), reorganizing tags and removing non-common tags (3 seconds). Then, inference for a test set of $100 - 200$ works takes about 10 to 15 seconds. Again, we will release all code upon acceptance, as we truly hope our tool can be of use to artists who are concerned by generative models potential infringing upon their unique styles.

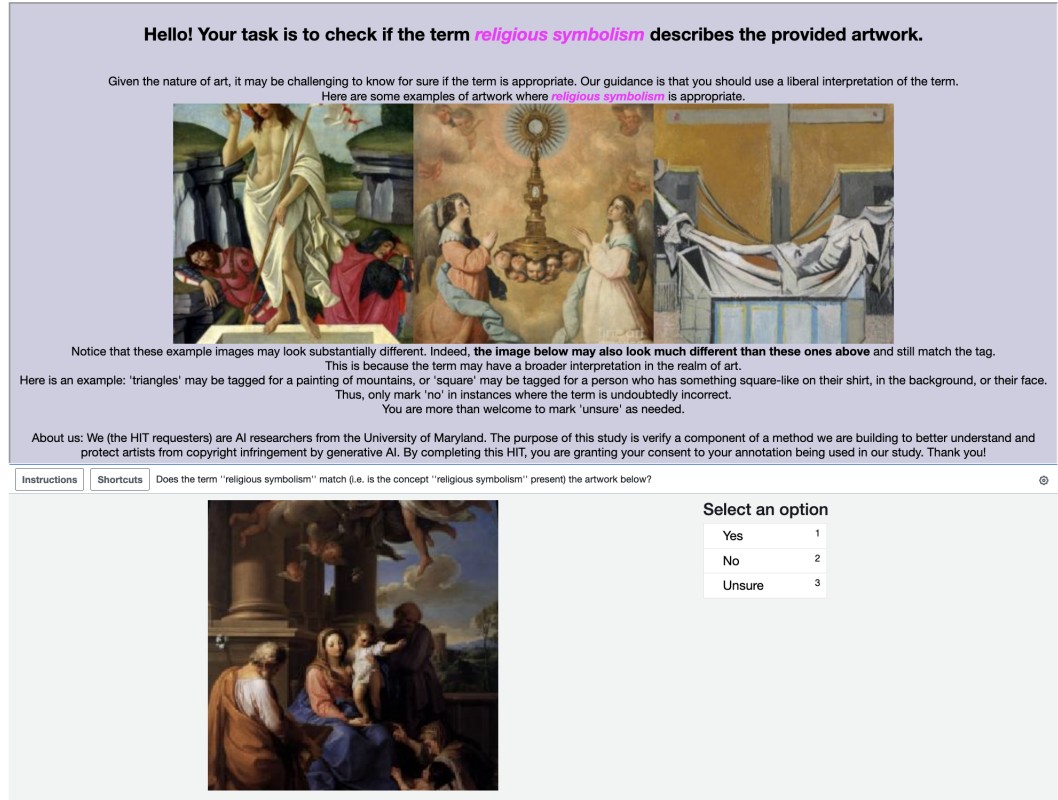

Figure 18: Instructions showed to MTurk workers to validate atomic tags.

## F.6 VALIDATION

Because tag match has multiple steps, we perform multiple validations. First, for image tagging, we utilize an MTurk study. We collect 3000 separate human judgements on instances of assigned atomic tags. Namely, we show 1000 randomly selected (tag, image) pairs to three annotators each. Figure 18 shows an example of the form presented to MTurk workers. MTurkers provide consent and are awarded $0.15 per task, resulting in an estimated hourly pay of $12 - $18. For each task, they answer 'yes', 'no', or 'unsure' to the question 'does the term {atomic tag} match the artwork below?' They are also shown example artworks for each term which were manually verified to be correct. Response rates were as follows: $69.89\%$ yes, $8.99\%$ unsure, $21.12\%$ no. In investigating inter-annotator agreement, we find that at least 2 annotators agree $92.1\%$ of the time, but all 3 agree only $51.52\%$ of the time. This reflects the subjectivity associated with assigning artistic tags, and partially motivates the need for a deterministic automated alternative, in order to objectively tag images at scale. All three annotators said no only $5.16\%$ of the time, and at least two said no $17.11\%$ of the time, suggesting that our zero-shot tagging mechanism achieves reasonable precision.

To validate the value of tag composition, we refer to figure 6, which shows how tags become more unique as they get longer (i.e. consist of more atomic tags). Moreover, our time analyses show that the added benefit of composing tags to find unique tag signatures does not come at the cost of the efficiency of our method. Finally, the non-trivial top-1 matching accuracy and strong top-5 matching accuracy shows that the extracted tag signatures do indeed capture some unique properties of artistic style. Figure 16 reflects a few more examples of successful inference, interpretation, and attribution for the task of detecting style copying by generative models.

|  |  | Top 1 | Top 5 | Top 10 |
|---|---|---|---|---|
| Generated Art | CompVis Stable Diffusion v1.4 | 10.10 | 35.49 | 49.74 |
|  | Stability AI Stable Diffusion v2 | 12.95 | 37.82 | 52.59 |
|  | PromptHero Openjourney | 6.99 | 31.87 | 45.08 |
|  | Average | 10.02 | 35.06 | 49.14 |
| Real Art (held out) |  | 61.56 | 82.53 | 88.44 |

Table 2: Match rates using TagMatch for three generative models, as well as on real held out art.

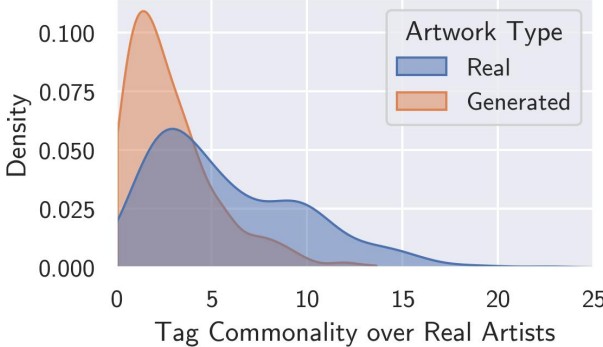

Figure 19: The tags for generated images are less common compared to tags in real art.

## G A SIM2REAL GAP IN TAG DISTRIBUTIONS

An added advantage of ascribing tags to images is that we can better compare image distributions from an interpretable basis (the tags). We briefly explore this direction now.

First, we provide complete results from applying TagMatch to generated images from each of the three text-to-image models in our study, presented in table 2. Consistent with our DeepMatch results, we observe substantially lower matching accuracy for generated images than for real held-out artwork. While the primary takeaway is that for many artists, generative models struggle to replicate their styles, we can also hypothesize that generative models may output images that follow a different distribution than the distribution of real artworks.

Motivated by this hypothesis, we now compare the distribution of real to generated artworks from the perspective of tags. Because we consider composed tags, the total space of tags is vast and hard to reason over. However, we can look at properties of each tags. Namely, we can inspect the uniqueness of tags. That is, for each tag present in generated images, we inspect the number of reference artists that also present that tag; we do the same for real art as well (subtracting one so to not double count the artist for which a given a tag is being considered). Figure 19 shows a kernel density estimation plot of the distributions of tag commonality, where a tag commonality of 5 means that for each tag assigned to a set of images (either from a real artist or from a generative model emulating an artist), 5 other artists also commonly use that tag. We see tags tend to be rather unique (due to our tag composition), and notably, tags for generated images are more unique.

## H HUMAN VALIDATION OF STYLE COPYING

We design and conduct a human study to assess style copying, toward verifying the outputs of our system. Given a 'plaintiff' artist A, we present a set of artworks generated by Stable Diffusion v1.4 in the style of the plaintiff A, along with two sets of real artworks: one from the plaintiff A, and another from a very similar artist B (selected as the artist – aside from A – that DeepMatch predicts to have created the highest number of the artworks generated to be in the style of A).

A human then denotes if the set of generated art more closely resembles the style of set from artist A or artist B, with the choice to abstain if neither set is more similar to the generated art than the other. Abstention may occur if the generated art is different from both sets or similar to both sets. Each set contains 16 works. The 16 generated artworks are selected at random from the collection of images generated in artist A's style. The 16 images each from artist A and B are drawn to be those that are most similar (via mean CLIP similarity) to the 16 selected generated images. The two sets of real art are presented in random order so that the human does not know which belongs to the plaintiff artist.

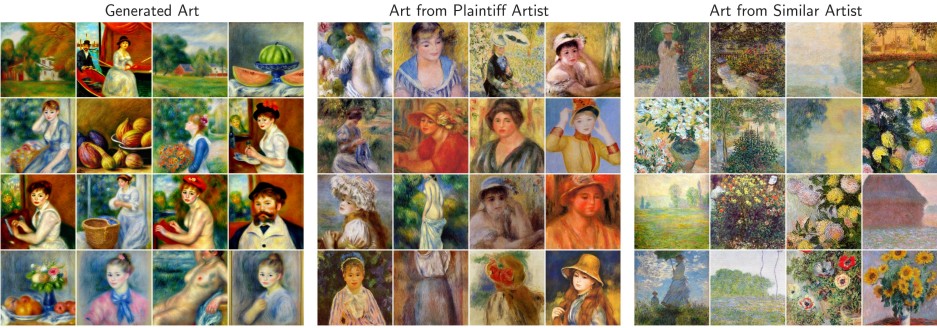

(a) Case 1: Generated art from *Pierre Auguste Renoir* was deemed more similar to art by Renoir than art from a similar artist. Here, the generated art seems to replicate aspects of Renoir's style.

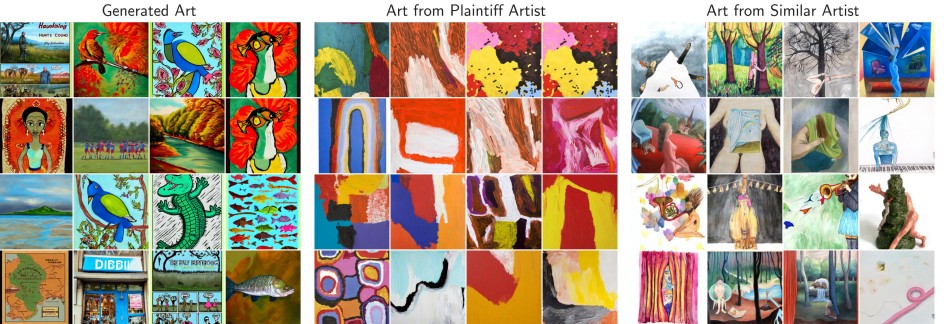

(b) Case 2: Generated art from *Sally Gabori* was deemed dissimilar to both real art from Gabori and another artist. Here, the generated art does not seem to be aware of Gabori's style.

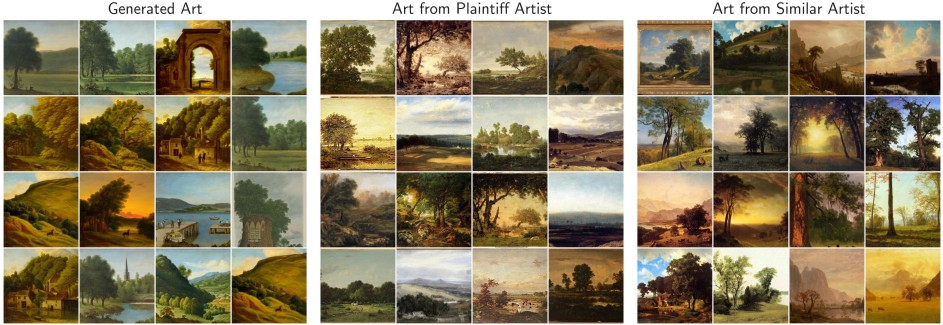

(c) Case 3: Generated art from *Theodore Rousseau* was deemed similar to both real art from Rousseau and from another artist, leading the human annotator to abstain from saying that the generated art copies Rousseau in particular. This highlights how style similarity may not constitute infringement.

Figure 20: Samples visualizations from our human evaluation. In the real evaluation, the order of the Plaintiff and Similar Artist panels is randomized, and the titles of the panels are removed. The first case demonstrates an instance where the human annotator viewed the generated art as more stylistically similar to the plaintiff's art than the most similar other artist, suggesting that Renoir's unique style is at risk of copying. The latter two cases show different ways in which the threshold of style copying may not be met: when the generated art is not similar to the plaintiff, or when the generated art is not sufficiently *uniquely* similar to the plaintiff.

We run this experiment for 40 artists: 20 which are flagged by our method as being at risk of style copying, and 20 which are not – we call the first group SAFE and the second group COPIED. We find that **the percent of artists where the generated art is marked as more similar to the plaintiff is** $90\%$ **for the COPIED group and just** $5\%$ **for the SAFE group**. Notably, the human abstained in 19 of the 20 artists in the SAFE group: for 15 artists, the generated art was dissimilar to both sets of real art, while for 4 artists, the generated art was equally similar to both sets. The latter type of abstentions underscores an important point: generated art can be similar to a plaintiff artist without necessarily meeting a bar for infringement, since infringement would require that a unique style is copied. This point relates to the discussion of CSD: while an improved style similarity metric can help surface relevant art, on its own, it does not immediately lead to an understanding of if an artist's style is unique, and what elements comprise that style. These questions are critical when it comes to copyright protection for artistic styles, and we center answering them in our work.

In summary, our human study corroborates the outputs of our method. Artists that are flagged to be at risk of style copying have generated artwork that is more similar to their work than even art from the most similar other artist. However, artists who are not flagged by our method have generated artwork that is either dissimilar to their style, or no more similar to their style than to that of another artist.

## I PATCH MATCH: GENERATING ADDITIONAL VISUAL EVIDENCE OF COPYING

Detecting artistic style copying in a given art requires analyzing local stylistic elements that manifest across an artist's body of work. To address this, we employ a patch-based approach that compares small image regions between a given art and original artworks, enabling a fine-grained analysis of stylistic and semantic (e.g. objects) similarities at a local level. We consider three patch matching methods: CLIP-based, DINO-based, and Gram matrix-based.

**Gram Matrix-based Patch Matching Gatys et al. (2016)**: The Gram matrix is a measure of style similarity introduced in the context of neural style transfer. It captures the correlations between the activations of different feature maps in a convolutional neural network, representing the style of an image. For patch matching, the Gram matrices of patches from the given art and original arts can be computed and compared using a suitable distance metric (e.g., Frobenius norm). The Gram matrix is specifically designed to capture stylistic elements, making it well-suited for detecting style copying.

**CLIP-based Patch Matching Radford et al. (2021)**: CLIP (Contrastive Language-Image Pre-training) is a powerful model that can effectively capture the semantic similarity between text and images. In the context of patch matching, CLIP embeddings can be used to measure the similarity between a patch from a given art and patches from original artworks. The patches can be encoded using the CLIP image encoder, and the cosine similarity between their embeddings can be computed to find the closest matches. CLIP may not be as sensitive to low-level stylistic elements, such as brushstrokes, textures, and color palettes, however it focuses more on higher-level semantic concepts, which can be useful to find if the given art pictured the same objects as the selected original patch.

**DINO-based Patch Matching Caron et al. (2021b)**: DINO is a self-supervised vision transformer that learns robust visual representations by solving a self-distillation task. DINO embeddings can be used for patch matching by computing the cosine similarity between the embeddings of patches from the given art and original artworks. We use DINO to capture higher semantical similarities, and check whether the given art pictured similar subjects of interest and high-level visual features as selected original artworks.

### I.1 EXPERIMENTAL SETTING

For our experiments, we aim to identify the most similar artwork from a pool of $10,000$ original artworks in the WikiArt dataset given a reference image. The reference image is first resized to a resolution of $512 * 512$ pixels and normalized. From this normalized image, we select a patch size of $128 * 128$ pixels. This process is repeated for all original artworks in the dataset, resulting in a total of $40,000$ patches from original artworks for comparison with the reference patch. We then use three methods, namely Gram matrix, CLIP, and DINO, to find the most similar patches.

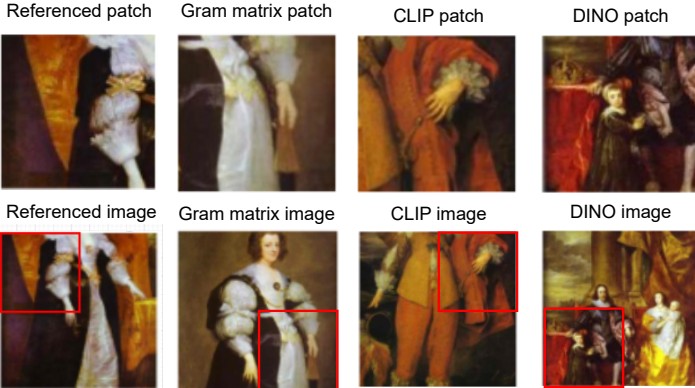

Figure 21: The most similar patches to a referenced patch in an image using Gram-matrix, CLIP, and DINO.

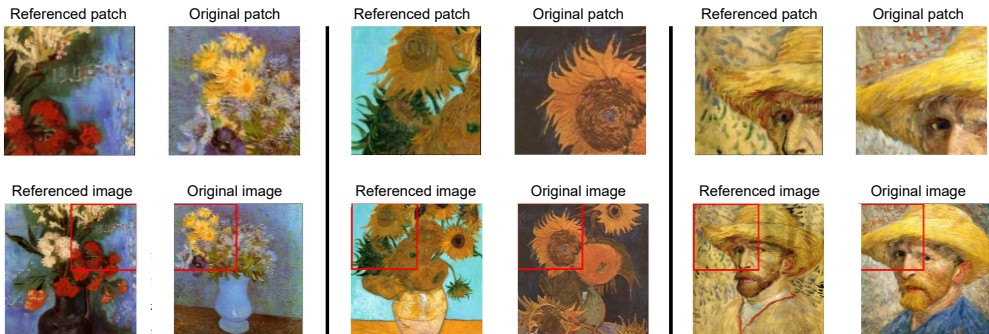

Figure 22: Comparison of patches using the Gram-matrix method, highlighting the closest matches to three selected artworks by Van Gogh. The selected original arts, all from Van Gogh, closely resemble the style of the referenced paintings.

Figure 21 showcases the patches that are deemed most similar to the image being referenced. These matches are determined using Gram-matrix, CLIP, and DINO methods.

We then select an artist and find patches from our original image dataset that closely match this artist's style. In Figure 22, we utilize the Gram-matrix method to identify the most similar patches to three chosen artworks by Van Gogh. Our dataset includes all paintings by Van Gogh as well as works by nine other artists. Gram-matrix selects original artworks that closely resemble the style of the reference image, all of which are from Van Gogh. Essentially, this means that Gram-matrix predominantly selects Van Gogh's artworks because they are the most stylistically similar to the referenced paintings compared to the works of the other nine artists.

## I.2 Discussion and limitations

Patch matching methods like Gram-matrix, CLIP, and DINO are effective in detecting similarities between artworks by examining their local stylistic and semantic elements. Gram-matrix focuses on capturing stylistic correlations, CLIP evaluates semantic similarity, and DINO concentrates on higher-level features. However, these methods have limitations. They primarily focus on local aspects of artworks and may overlook broader artistic characteristics such as texture, composition, and brushwork that are crucial to detect copyright infringements. Moreover, the process of finding the most similar patches for each given art takes approximately fifteen minutes when considering

$10,000$ original artworks, and if we opt to include more original artworks, the duration of the process would inevitably increase. Therefore, patch-matching methods are computationally expensive, which restricts their practical application. Despite these limitations, patch matching is valuable for identifying instances of direct copying in artworks and they aid in the detection of plagiarized content.

## J    DETAILS ON WIKIART SCRAPING

WikiArt is a free project intended to collect art from various institutions, like museums and universities, to make them readily accessible to a broader audience. We design a scraper to collect a corpus of reference artists, with which we can define a test artist's style in contrast to the other artists, and to provide a testbed to empirically study copying behavior of generative models. Some important landing pages to perform scraping are (i) the works by artist page (`https://www.wikiart.org/en/Alphabet/j/text-list`; url shows all artists starting with the letter 'j', and we loop through all letters), (ii) the page containing information on allowed usage (`https://www.wikiart.org/en/terms-of-use`), (iii) an example artist landing page (`https://www.wikiart.org/en/vincent-van-gogh`), and (iv) an example painting landing page (`https://www.wikiart.org/en/vincent-van-gogh/the-starry-night-1889`). As you can see, many pages have standard formats, making scraping particularly feasible. We will provide our scraping code, along with all other code, to facilitate easy updating of our dataset as time goes by.

We obtain artworks only from artists with at least 100 works on WikiArt, so to focus on somewhat famous artists who are arguably more likely to be copied. For every work, we also scrape the licensing information, and annotation for styles, genres, and title. In total, our dataset has 90,960 artworks over 372 artists. There are 81 styles with at least 100 works, with the most popular styles being *realism, impressionism, romanticism,* and *expressionism.* There were 37 genres with at least 100 works, with the most popular being *portrait, landscape, religious painting, sketch and study*, and *cityscape*. We note that we only include images who's license is either public domain or fair use, with the vast majority of works being public domain. Nonetheless, we strongly advise against using this dataset for commercial purposes, and especially for the purpose of copying artists.

