# OpenReview forum: "Rethinking Artistic Copyright Infringements In the Era Of Text-to-Image Generative Models"
_ICLR.cc/2025/Conference — ICLR 2025 Poster_

### Official Review · Reviewer_yjz4 · 2024-10-27

**Soundness:** 3
**Presentation:** 4
**Contribution:** 2
**Rating:** 6
**Confidence:** 4

**Summary:**

The paper aims to develop an “intuitive, automatic, legally-grounded” approach to determine style infringement. To do this, it trains two classifiers: one “DeepMatch,” a traditional image classifier trained to classify 372 artists based on a WikiArt training set, and a second, “TagMatch,” which classifies artist styles  using a human-interpretable tag-matching heuristic on top of more than 100 CLIP-derived tags across 10 aspects of artistic styles. Finally it conducts a measurement of images generated by popular diffusion models to quantify the number that resemble an artist’s style according to DeepMatch, and generates some explanations using TagMatch.

**Strengths:**

The problem area is important, with image synthesis models creating potentially huge economic impacts for the artistic professions. There is a need for scientific analysis to help guide discussions about implications for copyright and copyright law. Quantifying the amount of style imitation in the large models is a worthy goal. And in developing its methods, the paper recognizes the importance of interpretable explanations when it comes to human arguments.

**Weaknesses:**

[I have revised this score upwards based on revisions and discussions during rebuttal period; these are concerns having to do with the original submission]

The paper is not ready for publication in ICLR.  There are several specific problems

1. The legal discussion is not well-supported, and it is not sufficiently pedagogical for this audience.
2. The suitability of the classifiers for the task is not sufficiently measured or justified or compared to previous technical work.
3. The evaluation of style imitation in diffusion models does not support its conclusions.

The legal discussion is the most serious problem. For the technical audience at ICLR, a paper discussing legal issues must play a tutorial role. The audience at the conference is technical and not made of legal experts, so when making claims about legal issues, it is especially important for information about legal reasoning to be representative, explanatory, and correct. In contrast, this paper seems to be advancing an adventurous legal viewpoint in a venue with a legally unsophisticated audience.

Specifically, in the section on the legal framework, the paper puts in boldface: “**the task of showing the existence and uniqueness of artistic styles can be reduced to classification** – something deep networks are particularly adept at doing.” That claim appears to contradict the paper’s own legal citations. For example, when contemplating legal tests for style infringement, the cited work by Sobel makes a distinction between “extrinsic” similarity that can be described in words as opposed to an “intrinsic similarity” which is perceived by humans but not easily described in a mechanical way. Sobel illustrates the distinction by surveying many subtle situations that have appeared in case law. In the perspective provided by Adobe’s proposed style infringement legislation, the test is not pinned on the output, but rather, the intent of the AI user is placed at the center of style infringement. Both of these legal perspectives seem to be at odds with paper’s proposed reduction of the style infringement test to an automated and mechanical artist-identification image classification problem. Neither of these central legal issues are surfaced to the ICLR reader: the paper omits any contemplation, measurement, or comparison to the intrinsic judgements that would need to be made by a jury, nor does it make any measurement, prediction, or discussion of intent by the user of the AI.

This reviewer strongly feels that ICLR should not be the place to advance a new legal theory. Plenty of scientific questions arise in the legal discussions, such as whether automated methods might be able to anticipate the judgement of a jury (and if not, why not), or whether the intent of the user can be correctly guessed by an automated method.  At ICLR it would be more appropriate for the paper to pose and investigate a scientific question, and should not lead with a novel legal theory in boldface.



On the suitability of the classifier. More careful comparisons to previous work are needed. Several previous works have focused on the style classification such as Karayev and van Noord cited in footnote 2. However, the current work does not attempt to compare its approaches to any previous approaches, and it does not build on top of any of the evaluation approaches. For example van Noord takes on artist identification using the “Rijksmuseum Challenge” and analyzes and breaks down failed predictions. Do the proposed classifiers work better or worse than van Noord? Do they fail in similar ways? What is different in the technical approach that might lead us to expect the classifiers are more suitable?  Another insufficient comparison is between TagMatch and Concept Bottleneck Models. Table 1 in the appendix does a single pair of comparisons but does not quantify the sparsity advantage of TagMatch, or give any systematic comparison of meaningfulness to humans.  The heuristic in TagMatch seems ad-hoc: why would we expect its sparse set of labels to be more meaningful to a jury than the ones provided by CBM? No evaluation on that is done.



On the evaluation of existing style copying.  The paper’s conclusions are not sufficiently supported. The paper’s analysis of output from Stable Diffusion and OpenJourney concludes that most of the artist styles are not copied accurately, identifying just 16 of 372 artists whose styles are copied strongly.  However, no triangulation is done on this measurement, so it is unclear whether the low rate of identification is due to a weakness in the classifier, or whether it is due to a lack of style-imitation in the diffusion models.  A human evaluation on generated art style imitation could be done to check the estimate.  Or larger-scale data resources could be used, for example, the “parrotzone.art” project has identified several thousand styles that SD copies well, and these could potentially be used as an independent source of human assessment of style similarity.

**Questions:**

Is the goal of the paper to create an automated system to make quick judgements about style infringement without requiring humans to look?

Does your goal contradict Sobel's view that these judgments are not possible to articulate in clear categories, and that they are inherently the province of a human jury to make?

To what extent do you believe that your system matches the style-infringement judgements that a jury would make "by hand"?

What kinds of failure cases does the system have? What patterns characterize these failures?

If another scientist wishes to improve upon your system, what measurement can they do, that would indicate that they have made an improved system?

---

> ### Author Response · Authors · 2024-11-23
> **New appendix section to provide greater legal context**
>
> Thank you for your valuable feedback. We first address your concern the level of *legal discussion*.
>
> We completely agree that legal context is very important, as ICLR’s audience may have less exposure to relevant legal discourse. **To provide greater legal context in a more pedagogical manner, we add a new appendix section (B; after limitations)** that reviews the history of copyright (from concerns with printing presses in 1710 to chromolithography in 1870 to motion pictures in 1912 and sound recordings in 1971), the current legal landscape, and how our work may fit into it. Further, we clarify that we do not seek to ‘advance a new legal theory’ nor to simplify (via automation) the nuanced issue of copyright. We highlight the following paragraph from our new appendix section:
> > Our purpose in this paper is not to take a position on the issue of whether copyright law should be extended to protect individual artistic styles... *Neither is our purpose to automate the analysis of copyright infringement*. Rather, we are interested in investigating whether there is any scientific support for the idea that there are identifiable individual artistic styles, and that those styles could be correlated with a group of human-understandable stylistic terms. …when we try to train a model that can correctly identify authors of previously unseen works, we may get closer to understanding whether and when individual artistic styles exist. If we can link that classification to a selection of terms describing characteristics of those works, then we can explain to human beings what the components of such individual artistic styles might be. *If some kind of protection of individual style ever became part of copyright law, judges or juries would still have to decide whether a particular output of a generative AI tool too closely mimicked the individual style of an artist.  The degree to which an AI model could or could not correctly identify the author of a work, and the stylistic terms that that model could or could not correlate with that classification, would simply provide additional information to those human decision makers.*
>
> In other words, **our tool aims to assist human decision making, not replace it**. We identify how deep learning techniques can be used to quantitatively study relevant questions to the copyright debate, like if unique styles exist, how they can be articulated, and how much they reappear in AI-generated work.
>
> We specifically designed our methods so that jurors can engage with their outputs, even as our tool processes tens of thousands of artworks. Namely, **our tool can aid the ‘intrinsic’ decisions by**:
> - *Surfacing the most similar artists (and their works) to a plaintiff artist*, by seeing whose work is most frequently misclassified to the plaintiff or vice versa. The humans can then decide if the plaintiff’s style differs from the most similar artists sufficiently to have a 'unique' style. We explore this Appendix C.1 for instances where artists whose styles do not meet our threshold for uniqueness.
> - *Presenting side-by-side sets of art that match a tag signature, with the elements comprising that style articulated* via TagMatch’s interpretability and data attribution. The humans can compare the similarity of the two sets (do they both match the listed tags?) and compare these sets to artwork matching a subset of the tags (does adding one more atomic tag actually make the style more unique than other existing work?)
> - Engaging questions like ‘unprecedented similarity’, where *humans can inspect a plaintiff’s work, generated art, and art from the most similar existing artist*. Then, humans can decide if the generated art meets an unprecedented level of similarity to the plaintiff’s style, exceeding the similarity seen by any other human artist. We explore this deeper in Appendix C.2.
> Our tool uses the scalability of machines to gain insight and retrieve the most relevant artworks from a vast corpus, so that human decision makers can be more informed in their judgments. Also, we present these insights in an easy-to-digest manner: instead of saying Artist A and generated art A’ have a similarity of X (via some black-box metric), our tool says generated art A’ is confused as having been authored by artist A (over hundreds of other artists) Y% of the time, while also surfacing examples and articulating stylistic terms that contribute to the similarity.
>
> Intent of the AI user: we exclude this from our framework, as determining a user’s intent may be infeasible – we instead focus on questions with greater potential for quantitative approaches. We mention the Adobe work to show the emerging importance of artistic style copyright to legal and technical stakeholders alike.
>
> We'd be happy to discuss this topic further, as we agree that it is very important. In fact, a legal expert has been an integral part of our team from the start, so to appropriately ground our work in existing legal scholarship.

---

> > ### Author Response · Authors · 2024-11-23
> > **Comparisons to prior work -- novelty is in the framework and application of the components -- and answers to questions**
> >
> > **On comparison to prior work.** We emphasize that our central contribution is our framework – leveraging classification over image sets to quantify the uniqueness of an artistic style and the degree to which it is copied – more so than the implementation of it. This is particularly true for DeepMatch. As noted by both the reviewer and ourselves in the paper, others have trained art classifiers before, but these classifiers have not been used to study whether or not unique artistic styles exist and if they are copied, with an explicit goal of aiding legal decisions. This goal manifests in our prioritization of ease-of-use (our classifiers train in minutes) and interpretability (via TagMatch), which previous art classifiers lack. Nonetheless, we compare to multiple alternate implementations in Appendix C. Importantly, while individual accuracy numbers vary slightly, the relative ‘recognizability’ (accuracy per artist) is highly correlated across different classifiers. This leads us to believe that – to your question – ‘models fail in similar ways’, though we’d interpret low classification accuracy for an artist not as a ‘failure’, but instead as a signal suggesting that the artist’s style is not particularly unique.
> >
> > As for comparing the interpretability of TagMatch with CBMs, a human study would be required, which is out of scope, as this paper’s focus is its framework and findings, instead of any individual component. However, our claim on the potentially enhanced interpretability of TagMatch is based on prior human studies that show the importance of concise explanations in order for users to find them helpful: [1] argue a strict upper bound of 32 concepts before humans can no longer make use of them. The CBM baseline’s interpretability consists of similarity coefficients for 260 concepts. Making this 20% sparse with an l1 penalty drops accuracy, and still requires inspection of over 50 concept similarities, which may be cumbersome for a jury. In contrast, our method’s tag signatures are no longer than ten tags long, and also come with visual evidence due to TagMatch’s inherent attributions.
> >
> > [1] Ramaswamy et al, Overlooked Factors in Concept-Based Explanations: Dataset Choice, Concept Learnability, and Human Capability, CVPR ‘23
> >
> > **Answers to questions**:
> > - **The goal is not to create an automated system for copyright judgments so that humans do not need to look. We specifically design our system so that humans have more to look at, so that they can make more informed judgments**. We believe our goal – aiding humans in what is a very nuanced question – is directly in line with and inspired by Sobel’s scholarship on what factors (extrinsic/analytic and intrinsic/holistic) are to be considered in determining copyright infringement.
> > - We cannot answer the question of how aligned our tool’s outputs would be with judgments from actual juries because of the novelty of the issue at hand. **There is very limited, if any, precedent for decisions on artistic style, so we do not have sufficient ground truth to compare against**. Thus, there is no clear criteria that would define a ‘failure’ of our tool.
> > - One next step is doing human studies to gain insight as to how do real-life jurors feel about our tool and how the tool can be improved to better help jurors in making their decisions. TagMatch also opens up a line of research in interpretable and attributable by design classifiers, which can be studied in greater depth.
> >
> > Lastly, we note that **we are in the midst of conducting a human study to further validate our assessments of style copying**. Thank you for this suggestion! We will report our results as soon as we obtain them.
> >
> > Looking forward to your feedback on our initial rebuttals. And apologies for the late Friday night posting -- drafting our legal history took a bit of time.

---

> ### Comment · Reviewer_yjz4 · 2024-11-25
> **Main paper should clarify**
>
> Thanks to the authors for the addition of Appendix B. This discussion is much clearer than the framing in the main text!
>
> The appendix now clarifies (1) the articulation of the goal of "investigating whether there is any scientific support for the idea that there are identifiable individual artistic styles," and the (2) explicit clarification that "neither is our purpose to automate the analysis of copyright infringement."
>
> Given the clarified goals of the paper in the appendix, I suggest that the authors consider revising the paper to clarify (and pursue) this these points in the main text rather than just the appendix.
>
> Specifically: On lines ~87 the argument that the "existence and uniqueness of artistic styles can be reduced to classification" sticks out as an undefended assertion (empirically).  This equivalence of style to artist classification is not obvious to a computer vision expert - classifiers can have many inductive biases different from style - and this equivalence it is not investigated empirically in this paper, even though the paper asserts that the system is useful for understanding style.
>
> For example: an artist classifier might achieve high accuracy by looking for an artist's signature in the corner of the image (or some other confounding feature) rather than making a judgment based on the holistic style.  In light of the clarified goals, the lack of any check of this assertion is one of the paper's main weaknesses, and should be fixed before publication.
>
> The problem could be addressed if the authors were able to do two things:
>
> (1) Change the wording around line 87 to clarify that, although legal arguments would suggest that styles can be reduced to artist identification, one of your goals in this paper is to, as the authors say, "investigate whether there is any scientific support" for this view, i.e.,g to empirically test whether an artist classifier can successfully identify an artists' style.
>
> (2) Then such an investigation should be done, i.e., conduct an evaluation of your classifier(s) to compare whether their classification judgments match what people perceive as "style" in an image.  I can see a few ways to do it.  One way to do this would be to conduct a human evaluation.  Another way is to compare your classifier to the prior state-of-the-art in style identification, the CSD method from Somepalli 2024, i.e., to see if the judgements of your classifier would match their style metric.  The Somepalli work is open-source and should be able to be done with an automatic evaluation.  Alternately, you could adopt the Somepalli method as the basis for your classifier, and then you'd be able to argue that the empirical evidence they collected would support your point.
>
> Look forward to the authors' perspectives on the suggestions above.

---

> > ### Comment · Reviewer_yjz4 · 2024-11-25
> > **On contextualizing with respect to prior work**
> >
> > In view of the clarified goals of the paper, one of the most important prior work is Somepalli 2024, which is the current state-of-the-art in creating an empirically grounded artistic style metric.
> >
> > A couple points:
> > - The citation for Somepalli 2024 should be corrected.  Although recent, it was presented at ECCV 2024 (not just a preprint).
> > - The contributions, approach, and relevance of that work should be expanded a bit more in the text, and compared and contrasted with the approach taken in the current paper.
> > - Ideally, the method should be compared directly with Somppalli 2024's CSD method.
> >
> > In particular, Somepalli 2024 asserts that datasets "like WikiArt are not large enough to train a good style feature extractor" - but the WikiArt approach is exactly what the current paper is attempting.  The merits of this assertion should be discussed a bit.

---

> > > ### Author Response · Authors · 2024-11-25
> > > **brief update: new comparison w CSD; human validation in progress**
> > >
> > > Thank you very much for your continued engagement in discussing our work! We appreciate the time you're taking and the specificity and constructiveness of your suggestions.We wanted to provide a brief update before adding another comment + incorporating more suggestions later.
> > >
> > > First, **our human validation is on-going**, and we'll share the results as soon as we have them. We hope to show more evidence that humans agree with instances where the style of generated art for an artist is deemed not sufficiently similar to real art by that artist, according to our method. We note this task is not very easy for humans, as they need to have a vast knowledge of existing art to contextualize and better assess if a matched style in generated art is *unique* to the potentially copied artist. Fortunately, our method can help surface relevant art to aide the human in making this decision, which we will use in the set up of our human eval.
> > >
> > > We have also added a **new experimental comparison** to image similarity methods, including CSD. Namely, in Appendix D.3, we compare the retrieval abilities of CLIP, CSD, and our method [by repurposing softmax-probabilities from DeepMatch's classifier given an image as that image's representation] in surfacing art by the same artist given a query artwork -- see our rebuttal to reviewer ZD9A and App. D.3 for details. Our method outperforms the two baselines, even though DeepMatch is not originally intended for retrieval. The key distinction is that our approach directly seeks to find features that distinguish styles between *individual* artists, where as CSD contrasts broader artistic styles (not specific to each artist; e.g. CSD is optimized to distinguish impressionism from cubism, but not necessarily Monet's impressionism from Manet's impressionism). For questions around copyright of unique artistic styles, we believe it is necessary for us to (a) take on this greater level of distinction / specificity in characterizing individual artistic styles (b) prioritize making the outputs of our work readily accessible to a non-technical audience that may comprise the human decision makers (judges, juries, etc) we aim to assist.
> > >
> > > Other key notes about CSD:
> > > - It **can be easily be incorporated into our framework** as the backbone instead of CLIP. This is because the focus of our work is in a sense broader than that of CSD. We center the legal questions around copyright of individual styles, and design our framework around principles of ease-of-understanding/use and accessibility to a non-technical audience (via our 'recognizability' logical argument and interpretable components like TagMatch). CSD is an excellent contribution, and the resultant style fine-tuned backbone can be used in place of CLIP in our framework, so to leverage their improved embeddings while still delivering assessments of style copying in an easy to digest manner. We'll add evidence of how CSD and our method can be integrated in a  later comment.
> > > - **Their training (including contrastive and self-supervised losses to fine-tune a large model) is far more intensive than ours**, which requires just a simple classification loss on a small number of trainable parameters.  Thus, **while WikiArt was small for their purposes, it is suitably sized for ours**.
> > >
> > > Lastly, we **are in the process of modifying the main text per your suggestions**, including changing wordings to specify our goal, underscore the nuance of this problem, highlight the legal context (with explicit references to our new appendix section).

---

> > > > ### Author Response · Authors · 2024-11-26
> > > > **New human evaluation and study of adapting CSD as the basis for the classifier**
> > > >
> > > > We now present **two more new analyses** (in addition to the last retrieval experiment directly comparing CLIP, CSD, and DeepMatch), directly following your suggestions.
> > > >
> > > > **Human validation of style copying**. We design and conduct a human study to assess style copying, toward verifying the outputs of our system. Given a ‘plaintiff’ artist A, we present a set of artworks generated by Stable Diffusion v1.4 in the style of the plaintiff A, along with two sets of real artworks: one from the plaintiff A, and another from a very similar artist B (selected as the artist – aside from A – that DeepMatch predicts to have created the highest number of the artworks generated 'in the style of A'). A human then denotes if the set of generated art more closely resembles the style of the set from artist A or artist B, with the choice to abstain if neither set is more similar to the generated art than the other. Abstention may occur if the generated art is different from both sets or equally similar to both sets. Each set contains 16 works. The two sets of real art are presented in random order so that the human does not know which belongs to the plaintiff artist.
> > > >
> > > > We run this experiment for 40 artists: 20 which are flagged by our method as being at risk of style copying, and 20 which are not – we call the first group COPIED and the second group SAFE. We find that **the percent of artists where the generated art is marked as more similar to the plaintiff is 90% for the COPIED group and just 5% for the SAFE group**. Notably, the human abstained in 19 of the 20 artists in the SAFE group: for 15 artists, the generated art was dissimilar to both sets of real art, while for 4 artists, the generated art was equally similar to both sets. The latter type of abstentions underscores an important point: **generated art can be similar to a plaintiff artist without necessarily meeting a bar for infringement, since infringement would require that a *unique* style is copied. This point relates to the discussion of CSD: while an improved style similarity metric can help surface relevant art, on its own, it does not immediately lead to an understanding of if an artist’s style is unique, and what elements comprise that style. These questions are critical when it comes to copyright protection for artistic styles, and we center answering them in our work.**
> > > >
> > > > In summary, **our new human study corroborates the outputs of our method**. Artists that are flagged to be at risk of style copying have generated artwork that is more similar to their work than even art from the most similar other artist. However, artists who are not flagged by our method have generated artwork that is either dissimilar to their style, or no more similar to their style than to that of another artist.
> > > >
> > > > **Adopting CSD as the basis of our classifier**. We swap CLIP with CSD as the backbone of DeepMatch’s classifier, adding new results in App. D.1. Notably, the recognizability – rate at which art is recognized as belonging to an artist – over artists is highly correlated between the two choices of backbone, both for real held-out art (Pearson r=0.82) and generated art (Pearson r=0.83). This underscores that our work and CSD have different (but complementary) underlying goals, so CSD can easily be integrated to our framework. Namely, CSD aims to obtain an improved style similarity metric, while we focus on finding *unique* artistic styles. Still, in our prior new rebuttal experiment on image retrieval [more related to CSD's goal], we find that the softmax probabilities of DeepMatch serve as a strong image embedding for the task of retrieving works by the same artist of a query image, beating both CLIP and CSD for WikiArt (see new App D.3), suggesting DeepMatch may have utility in measuring style similarity on the level of image-pairs, though again, our goal is distinct from that task.
> > > >
> > > > **Updates to writing**: We have made changes to the text to highlight new studies, including the human validation, detailed legal history section, and CSD comparison you suggest, along with clarifications of the purpose of our work (including but not limited to explicitly stating that we do not wish to replace humans in the intro, sec 2, and conclusion).
> > > >
> > > > **Other minor clarifications**:
> > > > 1. Our method flags roughly 20% of artists to be at risk of style copying (which amounts to ~75 artists out of 372, not 16 like you mention).
> > > > 2. We study a very simple prompting setting, where each prompt consists of just a painting name with the suffix “by {artist}”. A more involved prompting strategy could lead to greater degrees of style copying.

---

> > > > > ### Comment · Reviewer_yjz4 · 2024-11-26
> > > > > **Help navigating changes**
> > > > >
> > > > > Thanks - a user study validating that DeepMatch correlates with human judgment would address a couple of my key concerns.
> > > > >
> > > > > Minor request: can you help me find the results in the paper - I'm having trouble navigating the changes without redlines - at which lines or sections should I look for the writeup of the user study?
> > > > >
> > > > > I notice the new footnote in the introduction which is helpful.

---

> > > > > > ### Author Response · Authors · 2024-11-26
> > > > > >
> > > > > > Certainly :) appendix H starts on page 30. We'd be happy to answer any follow up questions on the user study here as well.

---

### Official Review · Reviewer_ZD9A · 2024-10-27

**Soundness:** 2
**Presentation:** 2
**Contribution:** 2
**Rating:** 6
**Confidence:** 4

**Summary:**

This paper explores a significant question of how GenAI might infringe upon the styles of individual artists and if legal frameworks could protect these styles. In particular, the author developed a tool, ArtSavant, to measure and quantify artistic style infringement. ArtSavant mainly utilizes two methods:
* DeepMatch: aneural network classifier to establish a recognizable "signature" for an artist's style based on images.
* TagMatch: An interpretable, tag-based method, which decomposes artworks into stylistic elements or "tags".

Their empirical results show that GenAI models have the potential to reproduce unique artistic styles, rasing copyright concerns.

**Strengths:**

1. The paper is well-written and addresses a timely, important problem relevant to today’s AI and creative industries. The authors provide a solid combination of qualitative and quantitative results that contribute valuable insights into the field.
2.  Considering “style” as a central focus is an innovative approach. By shifting from image-wise similarity detection to a style-based classification specific to individual artists, the paper redefines the task in a way that offers a deeper understanding of style infringement.
3. The paper also emphasizes interpretability through the TagMatch method, which is especially useful in legal contexts, where clarity on how stylistic similarities are identified can support arguments around style infringement.

**Weaknesses:**

1. Although I enjoyed reading this paper and find “style” to be an intriguing approach to this problem, I am concerned about the inherent ambiguity surrounding this concept. The paper assumes that “style” can be quantitatively defined and detected, yet style is fundamentally a qualitative and fluid concept, often shaped by subjective interpretation. Additionally, even in the real world, many artists have very similar “styles,” which complicates the notion of unique stylistic signatures.

2. I wonder how a similarity-based method would perform on this dataset (please correct me if I missed this comparison in the paper). Are there cases where the style-based method detects something that a similarity-based method does not, or vice versa? A direct comparison could provide clearer insights into the advantages and limitations of each approach.

3. Regarding TagMatch, I understand its goal of enhancing interpretability; however, I find it somewhat limited in scope. First, it’s a relatively naive approach in some respects, relying solely on zero-shot CLIP with predefined tags. Second, “style” implies something more subtle and nuanced than broad artistic categories. Even within the same category, there can be vast differences between artworks, so I’m unsure of TagMatch’s practical utility in capturing the deeper, unique aspects of an artist’s style.

**Questions:**

1. Could you provide more quantitative and qualitative discussions for similarity-based vs. style based method
2. would appreciate any further clarifications regarding my concerns about Weaknesses. And I am willing to raise my score if I find them convincing

---

> ### Author Response · Authors · 2024-11-25
> **highlight: new quantitative comparison of our method to image-sim methods, w intuitive explanation**
>
> Thank you for your insightful comments! We apologize for extenuating circumstances leading to delay in our response. We address comments and discuss a new experiment below.
>
> **Ambiguity of style**. You're right! Style is qualitative, and this precisely is what makes questions about copyright protection for styles very challenging. Our work helps clear up some ambiguity around style by taking a quantitative approach based on legal scholarship and an intuitive logical argument (i.e. an artist has a unique style if their work can be consistently recognized). With this framework and central idea, we study key relevant questions, leading to new evidence suggesting that many artists have unique (recognizable) styles, and that generative AI may copy some of these styles. Thus, despite the ambiguity of style infringement, it requires study, as some artists may already be at risk.
>
> We also find evidence of some artists with very similar styles – exactly as you mention! – using our method (Appendix B.1), demonstrating the utility of our framework. Other specific ways in which our tool can help human decision makers include (a) surfacing the most similar artists to a plaintiff artist (based on whose work is most frequently misclassified to the plaintiff or vice versa), along with generated art, so that a jury can have more evidence to decide if the AI generations pose an unprecedented level of style similarity; (b) presenting side-by-side sets of art that match a tag signature, with elements comprising that style articulated via TagMatch. In summary, our tool provides quantitative insight to key questions around style and style infringement, toward helping the human decision makers (judges, juries, lawyers) navigate the nuance/ambiguity you mention.
>
> **Deeper comparison to image-similarity methods**
>
> We now present a new experiment comparing our method to baselines like CLIP and CSD (a version of CLIP fine-tuned to better measure style similarity; Somepalli et al ECCV '24). We test how well each method can retrieve images from a fixed reference set of artworks based on similarity to a query artwork. We take the train/test split of our dataset serve as reference/query sets for retrieval. We count each retrieved artwork created by the same artist as the query as a true positive, and if created by a different artist as a false positive. As DeepMatch is not intended for retrieval (unlike the baselines), we repurpose it's classifier for retrieval by using the output softmax probabilities for an image as an encoding of that image (i.e. each image is represented by a 372-dimensional vector, where the $i^{th}$ element corresponds to the likelihood that the image is authored by artist i).
>
> In Fig. 11 of App. D.3 (new), we plot the distribution of AUROCs over the artists in our dataset for the baseline methods and DeepMatch. We find that CSD outperforms CLIP in this task (mean AUROC of 0.89 vs. 0.86), and DeepMatch performs best of all (mean AUROC 0.98).
>
> Intuitively, our improved performance can be linked to the original objective of the three methods considered. Image similarity methods are not equipped or intended to measure stylistic similarity, but rather only a general sense of similarity between two images. Even methods such as CSD which are specifically trained using a contrastive loss to be invariant to style preserving transformations are not aware of the components that constitute a \emph{unique} artistic style. That is, \emph{while CSD is trained to contrast general art styles (e.g. *impressionism* from *cubism*), it is not trained to contrast between two (potentially very similar) artists (e.g. ***Manet's** impressionism* vs. ***Monet's** impressionism*). From a copyright perspective, we are precisely interested in only those stylistic components that set an artist’s work apart from their peers - that is, a unique artistic signature. This naturally suggests a framework which analyzes style from the lens of image classification. Our method, DeepMatch, is trained to classify artworks and thus upweights stylistic features which are most unique and thus useful for classification.
>
> **Composing tags makes them more specific, better capturing the nuance required to distinguish artists**. Precisely as the you note, each individual tag is insufficient to define a unique style, as artworks sharing one tag can be vastly different from one another. This is why tag composition is necessary: we observe that while each individual tag does not define any artist’s unique style, combinations of tags can form signatures, where only one artist frequently uses all the tags in the combination together (see Fig. 6). While TagMatch employs a zero-shot multi-label tagging scheme with CLIP, the true novelty of it comes in this tag composition step and how classification is ultimately performed  (via look-up of tag signatures), which makes it interpretable and attributable by design.
>
> We'd love to answer any follow up Qs! Thank you.

---

> > ### Comment · Reviewer_ZD9A · 2024-11-25
> >
> > Thank you for the response and for incorporating the experiment. The inherent ambiguity of the question presents challenges and limitations. **I still have some concerns about whether style or art can truly be quantified and identified in this way, as it may be overly naive, especially in a legal setting.** However, I do appreciate that the authors have made a solid attempt. While the method may not fully resolve the question, it represents meaningful progress and provides interesting insights. As a result, I have updated my score accordingly.

---

### Official Review · Reviewer_HLbq · 2024-11-05

**Soundness:** 4
**Presentation:** 4
**Contribution:** 4
**Rating:** 8
**Confidence:** 3

**Summary:**

This paper introduces ArtSavant, an explainable classifier for identifying artistic style infringements in generated art. The proposed framework consists of DeepMatch, a black-box neural classifier, and TagMatch, an interpretable tag-based method, to quantify the uniqueness of an artist’s style and recognize if it appears in generated images. The central idea is that if an artist’s works are consistently recognizable, they contain a unique style that can be classified. The approach uses both holistic and analytic style comparisons. It combines CLIP embeddings and tagged stylistic elements to support style infringement claims in a legally relevant, interpretable manner.

**Strengths:**

1. TagMatch offers an interpretable method for identifying stylistic elements, making it particularly valuable in legal contexts where explainability is essential.

2. The paper includes a comprehensive evaluation of the proposed methods, including both quantitative and human evaluation.

**Weaknesses:**

1. TagMatch relies on LLMs to generate concept vocabularies, which may limit its effectiveness for less-known artists whose stylistic elements may not be well-covered in pretraining data. Could you show how TagMatch performs on less-known artists? If there are some gaps between known and well-known artists, I am curious if there is way to enhance the vocabulary to better capture these unique styles?

2. DeepMatch uses a back-box for detection. However, such black-box classifiers may  pick up on spurious details rather than genuine stylistic features. For example, if an artist always includes a certain animal in his art works, DeepMatch might use this feature to classify the style. Could you provide some evidence that DeepMatch’s classification is based on broader stylistic elements instead of just this minor feature?

3. The preliminary study uses DINO features, which might be limited in representing stylistic nuances. Could you explore using features that are specifically trained for style similarity [1] to compare with your method as a baseline? What is the pro and con for classifier based approach proposed in this paper and embedding based approach?


4. The authors noted that a new artist could easily retrain the detector to include their works for the DeepMatch approach, as it’s quite efficient. However, I’m curious about the potential impact on performance. Does retraining lead to issues like catastrophic forgetting of previously learned styles? It would be interesting to see a case study where the existing classifier is expanded to include new artists, observing how this affects both new and original classifications.

[1] Unsupervised Image Style Embeddings for Retrieval and Recognition Tasks

**Questions:**

1. In line 21, there should be a space between the method name and the next word.

2. How many training examples from one artist are required to reliably detect the style of that single artist in DeepMatch?

3. Do DeepMatch and TagMatch provide different predictions for certain examples? If so, in what situations does this occur, and what are the characteristics of these artworks that lead to differing predictions?

---

> ### Author Response · Authors · 2024-11-27
>
> Thank you very much for your time and valuable feedback! We answer questions below.
>
> **Performance for lesser known artists**. As detailed in our response to reviewer gWF2, we find very little difference in performance for TagMatch over artists grouped by popularity (measured based on web traffic to their Wikipedia page); over four quartiles, TagMatch accuracies fall within a 7% range. While there is little to no gap based on popularity of the artist, we nonetheless note that TagMatch is flexible in that both its vocabulary and tagger can be swapped out with analogs that better cover more niche styles as needed, giving the user greater ability to engage with and modify TagMatch as needed.
>
> **Spurious correlations**. This is a great point and further emphasizes the need for an interpretable component of ArtSavant. Importantly, **it is rather non-trivial to decide what constitutes a ‘spurious’ feature in this case, as any sort of pattern can arguably be seen as a stylistic motif**. With the interpretability of our approach, ideally humans can decide for themselves if articulated stylistic elements are valid or spurious. Having qualitatively inspected many cases, including in a recent human study for reviewer yjz4, we can say that there have not been any obvious spurious features being relied upon. Nonetheless, one could leverage input-attribution techniques to better understand how DeepMatch makes predictions.
>
> **Classifier vs. embedding based approach**. We have added a number of explorations of using/comparing to style-finetuned embeddings. In short, (i) these embeddings can serve as a drop in replacement to the CLIP embeddings in our framework, without much of a change in outputs (ii) in a retrieval task (retrieving art from the same artist given a query image), softmax probabilities from our classifier turn out to be more effective than both vanilla and style-finetuned embeddings, likely because *our classifier explicitly upweights information related to distinguishing artists (relevant to our objective of understanding when and why artistic styles are ‘unique’), while other embeddings capture much more general information about the image* (this is the key pro / con or distinction between the two approaches, in addition to the enhanced interpretability of our approach).
>
> **Retraining capacity**. An earlier iteration of our dataset contained about 10 more artists, and we saw accuracy of our classifier then was only slightly less (~1pp less) due to the higher total count. Thus, for our proposed usage of just adding art for the one artist at a time, we believe there is minimal risk of forgetting.
>
> **Answers to questions**.
> - Thanks for pointing out the typo - it is now fixed.
> - The lowest number of works per artist in our training set is 80, and even at this count, there is an artist whose work is recognized with 82% accuracy.
> - DeepMatch is more accurate, so TagMatch sometimes confuses stylistically similar artists when DeepMatch is correct, though generally, the predicted artist using DeepMatch is within the top 5 predicted artists using TagMatch. We can still inspect matched tags in these cases, allowing us to make use of TagMatch’s inherent interpretability and attributions.

---

### Official Review · Reviewer_gWF2 · 2024-11-09

**Soundness:** 3
**Presentation:** 3
**Contribution:** 3
**Rating:** 8
**Confidence:** 4

**Summary:**

The paper introduces ArtSavant, a tool designed to assess artistic style copying in text-to-image generative models. Built on legal scholarship, ArtSavant combines two methods, DeepMatch (a neural classifier) and TagMatch (an interpretable tag-based approach), to detect unique artistic styles and assess whether they are replicated in generated images. An empirical study using ArtSavant indicates that around 20% of the artists in their dataset appear at risk of style copying by generative models, raising concerns for the need to protect artistic styles under copyright.

**Strengths:**

1. The paper addresses a timely issue -- potential copyright infringements in text-to-image generation -- that bridges technical, legal, and ethical domains.
2. ArtSavant’s combination of DeepMatch and TagMatch represents a thoughtful approach, with one method offering high accuracy and the other interpretability. This approach is likely beneficial for non-technical audiences, such as legal professionals and artists.
3. The paper is well-grounded in legal discussions, positioning ArtSavant as a tool that can potentially support legal decision-making regarding style infringement.

**Weaknesses:**

1. The use of a limited reference dataset (372 artists) could affect the generalizability of ArtSavant’s findings, especially for artists with less established styles. Expanding the dataset to include more diverse artistic styles could strengthen the conclusions.
2. ArtSavant may struggle with assessing artists whose work doesn’t conform to traditional or well-known styles, limiting its broader applicability. It may inadvertently favor more mainstream artistic elements, possibly overlooking style copying for non-Western, niche, or experimental art styles.
3. Although TagMatch aims to make the tool interpretable, the subjectivity inherent in artistic tagging could affect its reliability, especially in legal contexts. This may be partially addressed by improving tagging accuracy, as noted by the authors.

**Questions:**

1. How does ArtSavant perform when applied to more obscure or emerging artists whose styles may be less distinctive or well-known?
2. The TagMatch method relies on zero-shot tagging with CLIP, which may not capture subtleties in artistic style. Have the authors considered evaluating the reliability of TagMatch across different art genres or complex styles, and could a more refined tagging approach improve interpretability and consistency?

---

> ### Author Response · Authors · 2024-11-26
> **Confirmed that biases are not present**
>
> Thank you for your kind words (especially on the topic of our legal discussions and the accessibility of our approach to broader audiences) and valuable feedback! We answer questions below. Sorry for the delay.
>
> **Checking for potential biases**. We appreciate the reviewer’s attention to this important issue. To check for biases more closely, we collect two new forms of metadata: we obtain artist **nationality** from WikiArt when available, and we proxy ‘**popularity**’ by counting the number of visits to each artist’s wikipedia page using PageMetrics (inspired by [1] who also used web traffic to proxy popularity). Then, we inspect performance of DeepMatch and TagMatch on real held-out art over different continents and popularity levels. Both are stable: for regions, DeepMatch average accuracies and TagMatch accuracies fall within a 6% and 3% range respectively. For popularity, over four equally sized quartiles, DeepMatch average accuracies and TagMatch accuracies fall within a 3% and 7% range.  Since we have an accuracy per artist for DeepMatch (whereas TagMatch either matches or does not match the artist), we can also inspect the correlation between our popularity measure and the DeepMatch confidence per artist: we find there to be virtually none (Pearson r of 0.04).
>
> **Diversity of dataset**. With our new nationality data, along with previously collected metadata, we were able to see that our dataset has artists from all 6 continents (aside from Antarctica) and at least 40 nations, while also spanning 81 distinct styles (according to WikiArt’s categorizations). Importantly,  our code (to be released) allows users to easily expand our dataset to artists with less than 100 artworks on wikiart – this threshold is somewhat arbitrary, and related more to our goal of measuring copying by generative models of ‘prolific’ artists.
>
> As for artists that fall under styles (based on WikiArt) that are more niche, we observe them to actually be recognized at higher rates, with a small to moderate positive correlation of r=0.4 between DeepMatch accuracy on held out art for an artist and the number of other artists who fall under the same style category as that artist. The intuition here is that if only one artist practices some style, their work will be more recognizable than an artist that practices a style that is very common, like Italian renaissance art (see Figs. 4 and 10).
>
> **TagMatch’s components (vocabulary, tagger) are flexible**. We note that TagMatch is sufficiently modular to where an improved tagger can easily be swapped in. More readily, one can modify the underlying vocabulary to incorporate more descriptors relevant to some broader style of interest. Thus, the reviewer is correct in that TagMatch can easily be improved as underlying technology evolves.
>
> [1] Sun et al, "Head-to-Tail: How Knowledgeable are Large Language Models (LLMs)? A.K.A. Will LLMs Replace Knowledge Graphs?", NAACL 2024

---

### Author Response · Authors · 2024-11-27
**Summary of updates**

First, thank you to all reviewers for their time and engagement -- it is privilege to get your continued feedback.

We would like to briefly summarize the main updates to the paper made during the discussion period thus far:
- **New extended discussion of legal context**: We have added App. B to serve as a crash-course / starting point for readers to understand the legal context of our work, going over centuries history and over a dozen cases.
- **New experiment directly comparing image-similarity methods to our approach**: We find that DeepMatch's softmax probabilities are surprisingly strong when used as image representations for the retrieval task of getting art from the same artist as a query image. We directly compare to (and beat) embeddings from CLIP and a recent style fine-tuned CLIP. We also stress that our work tackles a distinct (though related) problem to style similarity -- we are more concerned with finding and articulating *unique* styles, as we find this critical to the copyright discussion. Details in App D.3.
- **New human study validating our style copying judgments**: We design and carry out a user study to assess if human judgments match the outputs of our tool when determining if generated art infringes on an artist's unique style (App. H). The human study corroborates our automatic judgments, and sheds insight on how style similarity differs from our task of (unique) artistic style infringement.
- Other small highlights: like **demonstrating that style fine-tuned embeddings can be swapped into our method easily and with little change to results**, and **quantitative confirmation that our tool is not biased to popularity or geography**.

We look forward to the rest of the discussion period and would be more than happy to clear up any more concerns.

---

### Meta-Review · Area_Chair_N4CW · 2024-12-22

**Metareview:**

The authors propose to quantify the degree of artistic style infringement by measuring the extent to which an automated classifier can recognize the style of that artist. This is a nice, simple idea that attempts to tackle a highly topical and otherwise thorny problem, and I appreciate the authors' attempts to bring clarity and attention to this issue. The initial version of the paper was unclear about the legal positioning it was adopting as well as some baselines, but these issues were addressed in the rebuttal. After discussion, reviewers were unanimous in recommending acceptance and I am happy to recommend acceptance as well.

I encourage the authors to incorporate the reviewers' feedback into their camera-ready. In particular, please be careful about the legal claims made in the paper. They are significantly less controversial after revision, but I note that the revised introduction still suggests that, e.g., automatic quantitative approaches (and specifically, your particular automatic quantitative approach) could be used entirely to make a legal judgement.

**Additional Comments On Reviewer Discussion:**

The most substantive changes were around what Reviewer yjz4 suggested wrt legal framing and baselines. These were satisfactorily addressed in the rebuttal.

---

### Decision · Program_Chairs · 2025-01-22

Accept (Poster)